# PRE-TRAINING EPIDEMIC TIME SERIES FORECASTERS WITH COMPARTMENTAL PROTOTYPES

## ABSTRACT

Accurate epidemic forecasting is crucial for outbreak preparedness, but existing data-driven models are often brittle. Typically trained on a single pathogen, they struggle with data scarcity during new outbreaks and fail under distribution shifts caused by viral evolution or interventions. However, decades of surveillance data from diverse diseases offer an untapped source of transferable knowledge. To leverage the collective lessons from history, we propose CAPE, the first open-source pre-trained model for epidemic forecasting. Unlike existing time series foundation models that overlook epidemiological challenges, CAPE models epidemic dynamics as mixtures of latent population states, termed *compartmental prototypes*. It discovers a flexible dictionary of compartment prototypes directly from surveillance data, enabling each outbreak to be expressed as a time-varying mixture that links observed infections to latent population states. To promote robust generalization, CAPE combines self-supervised pre-training objectives with lightweight epidemic-aware regularizers that align the learned prototypes with epidemiological semantics. On a comprehensive benchmark spanning 17 diseases and 50+ regions, CAPE significantly outperforms strong baselines in zero-shot, few-shot, and full-shot forecasting. This work represents a principled step toward pre-trained epidemic models that are both transferable and epidemiologically grounded.

## 1 INTRODUCTION

Infectious disease outbreaks pose a persistent threat to global public health and economic stability (Nicola et al., 2020). Effective outbreak management relies on accurate epidemic forecasting—the prediction of future cases, hospitalizations, and other critical metrics (Liu et al., 2024b; Wan et al., 2024; Adhikari et al., 2019). A wide range of models have been developed to provide these crucial forecasts, which generally fall into two categories. **Mechanistic models**, such as the classic Susceptible-Infected-Recovered (SIR) (Cooper et al., 2020) approach, are grounded in epidemiological principles; they divide a population into *compartments* that represent distinct *population states* (e.g., susceptible, infectious, recovered) and use differential equations to explicitly model flows between these states. In contrast, modern **machine learning** methods like LSTMs (Shahid et al., 2020) learn complex patterns directly from historical data, offering greater flexibility without imposing a predefined structure.

However, these data-driven forecasters are often trained for a single pathogen in a specific region. This narrow scope makes them brittle: they face acute data scarcity during the critical early stages of a novel outbreak, and they fail under distribution shifts induced by viral evolution, behavioral change, or policy interventions. At the same time, decades of surveillance across diverse pathogens and geographies remain an untapped source of transferable structure. Motivated by the success of large pre-trained models in language, vision, and time-series domains (Zhao et al., 2023), we ask: *Can we build a large pre-trained epidemic forecaster that learns from the collective history of infectious diseases to improve generalization and robustness?*

Simply applying a general time series foundation model (Liang et al., 2024) is insufficient, as it overlooks core epidemiological challenges: (1) *Structural heterogeneity*: Pathogens follow different effective compartmental progressions (e.g., SIR vs. SEIR (He et al., 2020)), so a single fixed mechanism cannot transfer broadly across diseases and regions. (2) *Hidden population states*: Surveillance data records only reported infections, while important states such as exposure, susceptibility, and

immunity are not directly observed. (3) *Distribution Shifts*: Interventions, behavioral changes, and pathogen evolution induce abrupt non-stationarities, often when outbreak histories are shortest. These properties demand powerful epidemic pre-trained models that can adapt to diverse pathogens, disentangle hidden population states, and remain robust under shifts.

**Our Solution.** We introduce CAPE (CompArtment Pre-training for Epidemics), a pre-trained framework that learns epidemic dynamics as a mixture of latent population states, termed **compartmental prototypes**. (1) To address structural heterogeneity, rather than relying on a rigid, pre-defined compartmental structure, CAPE discovers a flexible dictionary of latent compartments directly from data. Each outbreak sequence is modeled as a mixture that varies in time in these prototypes, linking observed infections to latent population states. (2) To handle hidden drivers, the learned mixtures act as proxies for unobserved states such as susceptibility, disentangling latent population dynamics from noisy observed case counts. (3) To address distribution shifts, CAPE employs two self-supervised pre-training strategies to encourage robust representations that generalize under non-stationarities and scarce data. In addition, we further propose lightweight epidemic-aware regularizers to align learned prototypes with epidemiological semantics. Our contributions include:

(1) **Pre-training framework for epidemic time series forecasting:** We introduce the first open-source pre-training framework [1] for epidemic forecasting. It learns latent compartmental prototypes directly from time series, guided by several epidemic-aware losses that regularize the model's predictions and learned prototype representations.

(2) **Comprehensive benchmark for epidemic pre-training:** We assemble a diverse pre-training and evaluation suite, spanning 17 diseases across 50+ regions for pre-training and 5 downstream datasets covering 4 challenging settings (zero-shot, few-shot, cross-location, and cross-disease).

(3) **State-of-the-art performance in diverse forecasting settings:** We demonstrate the effectiveness of our pre-trained model, which significantly outperforms existing benchmarks by an average of 6.3% lower average MSE in the full-shot setting and 10.3% lower average MSE in the few-shot setting across all tested downstream datasets.

(4) **In-depth analysis:** We conduct extensive analyses to provide insights into how the learned latent prototypes improve forecasting accuracy and show that pre-training effectively learns the representation of diverse diseases and mitigates the impact of distribution shifts.

## 2 RELATED WORK AND PROBLEM DEFINITION

**Epidemic Forecasting Models.** Traditionally, epidemic forecasting employs models like ARIMA (Sahai et al., 2020), SEIR (He et al., 2020), and VAR (Shang et al., 2021). ARIMA predicts infections by analyzing past data and errors, SEIR models population transitions using differential equations, and VAR captures linear inter-dependencies by modeling each variable based on past values. Recently, deep learning models, categorized into RNN-based, MLP-based, and transformer-based, have surpassed these methods. RNN-based models like LSTM (Wang et al., 2020), GRU (Natarajan et al., 2023), and more epidemic-specific models like EpiDeep Adhikari et al. (2019) and EINNs Rodríguez et al. (2023) use gating mechanisms to manage information flow. MLP-based models use linear layers (Zeng et al., 2023) or multi-layer perceptrons (Borghi et al., 2021; Madden et al., 2024) for efficient data-to-prediction mapping and physics-informed distillation Wang et al. (2021). Transformer-based models (Wu et al., 2021; Zhou et al., 2021; 2022) apply self-attention to encode time series and generate predictions via a decoder. However, these models are limited as they typically utilize data from only one type of disease without considering valuable insights from diverse disease datasets.

**Pre-trained Time Series Models.** To enable few-shot or zero-shot capabilities, transformer-based models often employ pre-training on large datasets, which typically use masked data reconstruction (Zerveas et al., 2021; Rasul et al., 2023) or promote alignment across different contexts (Fraikin et al., 2023; Zhang et al., 2022; Yue et al., 2022). For example, PatchTST (Nie et al., 2022) segments time series into patches, masks some, and reconstructs the masked segments. Larger foundational models like MOMENT (Goswami et al., 2024) aim to excel in multiple tasks (e.g., forecasting, imputation, classification) but require substantial data and computational resources. In epidemic contexts, Kamarthi et al. (Kamarthi & Prakash, 2023) pre-train a model on various diseases, improving downstream performance and highlighting pre-training's potential in epidemic forecasting. However, the complete implementation is not publicly available. Moreover, existing approaches

---

[1] https://anonymous.4open.science/r/CAPE_ICLR26-A041/

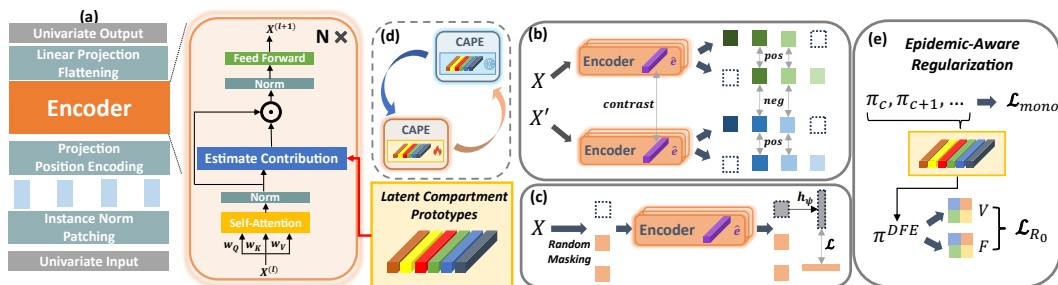

Figure 1: (a) CAPE encoder with latent compartment prototypes; (b) Hierarchical contrasting for temporal representations; (c) Random masking and reconstruction; (d) Optimizing the encoder and prototype representations alternatively. (e) Epidemic-aware regularization, including losses for monotonic and non-monotonic dynamics.

overlook hidden compartmental influence and zero-shot ability in epidemic forecasting and lack a deep analysis of how pre-training materials impact downstream performance. In this study, we introduce latent compartment modeling and conduct a thorough analysis of these questions (see A.6 for more discussions).

**Problem Definition.** Given a historical time series input: $\mathbf{x} \in \mathbb{R}^{T \times 1}$, where $T$ is the size of lookback window, the goal of epidemic forecasting is to map $\mathbf{x}$ into target trajectories (e.g. infection rates): $\mathbf{y} \in \mathbb{R}^h$, where $h$ denotes the size of the forecast horizon. We define $X$ and $Y$ as the random variables of input and target, respectively. During pre-training, a representation function $g_\theta : \mathbb{R}^{T \times 1} \to \mathbb{R}^{T \times d}$, where $d$ denotes the dimension of the latent space and $\theta$ being the parameter of the model, extracts universal properties from a large collection of epidemic time series datasets $\mathcal{D}_{\text{pre}} = \{D'_1, D'_2, \ldots, D'_S\}$. Then, a set of self-supervised tasks $\mathcal{T}_{\text{pre}} = \{\mathcal{T}_i\}_{i=1}^R$ is defined, and each $\mathcal{T}_i$ transforms a sample $\mathbf{x} \sim \mathcal{D}_{\text{pre}}$ into a new input-label pair: $(\tilde{\mathbf{x}}, \tilde{\mathbf{y}})$, and optimizes a loss $\mathcal{L}_{\mathcal{T}_i} = \mathbb{E}_{\mathbf{x} \sim \mathcal{D}_{\text{pre}}}[\ell_{\mathcal{T}_i}(h_\psi(g_\theta(\tilde{\mathbf{x}})), \tilde{\mathbf{y}})]$, with $\ell_{\mathcal{T}_i}$ being the task-specific metric and $h_\psi$ the task-specific head.

## 3 PROPOSED METHOD

Our pre-training framework is designed to overcome the core challenges of *structural heterogeneity*, *hidden drivers*, and *distribution shifts* inherent in epidemic forecasting. We address these issues through two main contributions: (1) a flexible model architecture that learns latent compartmental prototypes directly from observational data, and (2) a set of epidemic-aware pre-training objectives that guide the model to learn robust, generalizable representations. We will elaborate on these architectural and objective-based solutions in the following subsections.

### 3.1 MODELING LATENT COMPARTMENTAL PROTOTYPES

**Temporal Backbone.** Following the prior work on patch-based time series modeling (Nie et al., 2022), we segment the input sequence of infection counts $\mathbf{x}$ into non-overlapping temporal patches, $\mathbf{x} = [\mathbf{x}_1, \ldots, \mathbf{x}_C]$, where each patch $\mathbf{x}_c \in \mathbb{R}^{T/C}$. This patching strategy enables the model to capture local temporal patterns. A standard self-attention encoder, $f_{\text{enc}}$, then processes these patches to learn long-range temporal dependencies, producing a contextualized representation $\mathbf{h}_c^{(l)} = f_{enc}(\mathbf{x}_c^l)$ for each patch $c$ at a given layer $l$.

**Compartmental Prototypes Learning.** Different diseases exhibit different progression patterns: some show a simple rise-and-fall in cases, while others involve additional hidden stages such as incubation periods. In epidemiological terms, this corresponds to differences in compartmental structures (e.g., SIR (Cooper et al., 2020) vs. SEIR (He et al., 2020)). Epidemic forecasting is therefore challenged by *structural heterogeneity* and *hidden population states*: different structures generate diverse dynamics, while only a subset of compartments are directly observable. Classic compartmental models impose a rigid, pre-defined structure that cannot adapt to various types of diseases and scenarios. To address this, we move beyond fixed models and propose a framework that learns to represent epidemic dynamics as a *dynamic mixture of latent population states*, which we term **compartmental prototypes**. Our approach is analogous to learning a vocabulary of core epidemiological behaviors directly from data. Each outbreak is expressed as a time-varying mixture

over these prototypes. For example, during the growth phase of influenza, the model may infer that roughly 30% observed case increases are explained by infectious-like prototypes, while the rest reflects susceptible depletion and recovery. Formally, we initialize a set of $K$ learnable embeddings, $\mathbf{E} = \{\mathbf{e}_k\}_{k=1}^K \in \mathbb{R}^{K \times d}$, where each $\mathbf{e}_k$ is a "prototype" representing a basic compartment. For any given time window (a patch $c$), the model's crucial task is to determine the contributions of these prototypes in representing the current epidemic time series. We accomplishes this by inferring a set of mixture weights, $\boldsymbol{\pi}_c = [\pi_{1,c}, \dots, \pi_{K,c}]$, using a cross-attention mechanism between the patch's representation $\mathbf{h}_c^{(l)}$ and the full set of prototype embeddings $\mathbf{E}$:

$$\pi_{k,c}^{(l)} = \text{Softmax}\left( (\mathbf{W}_k^{(l)} \mathbf{e}_k)^\top \cdot (\mathbf{W}_s^{(l)} \mathbf{h}_c^{(l)}) \right), \tag{1}$$

where $\mathbf{W}_k^{(l)}$ and $\mathbf{W}_s^{(l)}$ are learnable linear projections. $\boldsymbol{\pi}_c$ quantifies the contribution of each compartmental prototype in representing the current patch, forming a regularized and robust representation for forecasting. The patch representation is then updated by taking a weighted sum over the Hadamard product ($\odot$) of the patch representation and the compartment embeddings. This allows the model to modulate the observed time series data with the inferred underlying dynamics. The layer-wise update is defined as:

$$\mathbf{x}_c^{(l+1)} = \sigma\left( \mathbf{W}_f^{(l)} \sum_{k=1}^K \pi_{k,c}^{(l)} \left[ f_{enc}(\mathbf{x}_c^{(l)}) \odot \mathbf{e}_k \right] \right), \tag{2}$$

where $\sigma$ represents a feed-forward block containing the projection $\mathbf{W}_f^{(l)}$. After stacking $L$ such layers, a final task-specific linear head, $h_\psi$, maps the resulting representations $\mathbf{x}^{(L)}$ to the target prediction $\hat{\mathbf{y}} = h_\psi(\mathbf{x}^{(L)})$.

## 3.2 SELF-SUPERVISED PRE-TRAINING

To learn robust representations that can withstand the *distribution shifts* and data scarcity common in epidemics, we employ two self-supervised pre-training objectives designed to capture universal patterns across diverse time series.

**Masked Time-Series Reconstruction.** We use a masked autoencoding task to teach the model the underlying grammar of epidemic curves. By randomly masking a fraction (e.g., 30%) of the input patches and training the model to reconstruct the original series, we force it to learn meaningful temporal interpolations. The objective is to minimize the Mean Squared Error, $\mathcal{L}_{\text{recon}} = \text{MSE}(\hat{\mathbf{x}}, \mathbf{x})$. This builds resilience to the noisy and incomplete data often encountered during chaotic outbreak periods, improving the model's fundamental forecasting capabilities.

**Contrastive Learning for Compartmental Prototypes.** A key challenge during distribution shifts is that epidemic curves can become highly non-stationary, and superficially similar patterns might arise from vastly different underlying dynamics. To prevent our model from learning spurious correlations, we introduce a contrastive objective that regularizes the compartmental prototype mechanism itself. The goal is to ensure that the inferred contributions of compartments ($\boldsymbol{\pi}_c$) are both consistent and discriminative. Specifically, we enforce two conditions: (1) two different augmented views of the same time-series patch should be mapped to a similar mixture of compartmental prototypes (positive pairs), and (2) patches from different, epidemiologically distinct contexts should be mapped to dissimilar compartmental prototypes (negative pairs). This pushes the model to focus on the essential, underlying dynamics captured by the prototypes, rather than overfitting to superficial noise. The patch-wise contrastive loss is defined as:

$$\mathcal{L}_{\text{CL}}(j,c) = -\mathbf{X}_{(j,c)} \cdot \mathbf{X}'_{(j,c)} + \log\left( \sum_{b \in B} \exp\left( \mathbf{X}_{(j,c)} \cdot \mathbf{X}'_{(b,c)} \right) + \mathbb{I}_{j \neq b} \exp\left( \mathbf{X}_{(j,c)} \cdot \mathbf{X}_{(b,c)} \right) \right)$$
$$+ \log\left( \sum_{t \in \Omega} \exp\left( \mathbf{X}_{(j,c)} \cdot \mathbf{X}'_{(j,t)} \right) + \mathbb{I}_{c \neq t} \exp\left( \mathbf{X}_{(j,c)} \cdot \mathbf{X}_{(j,t)} \right) \right), \tag{3}$$

where $B$ is the batch, $\Omega$ is the set of overlapping patches, and $\mathbb{I}$ is the indicator function.

## 3.3 EPIDEMIC-AWARE REGULARIZATION

To ensure our compartmental prototypes learn epidemiologically plausible dynamics, we introduce three regularization terms that instill prior knowledge from classic mechanistic models. These regularizers help disentangle the learned prototypes, encouraging them to represent distinct, interpretable dynamics (e.g., monotonic vs. non-monotonic). We apply these regularizers with a small weight

during pre-training to gently guide representation learning, and with a larger weight during fine-tuning to specialize the model to a specific pathogen.

**Monotonic Dynamics.** Certain compartments, such as Susceptible or Recovered, typically exhibit monotonic behavior (Nguyen et al., 2023). To enforce this, we introduce a monotonic loss. For a compartment $k$ with an expected monotonic trend, let $\boldsymbol{\pi}_k = [\pi_{k,1}^{(L)}, \ldots, \pi_{k,C}^{(L)}]$ be its mixture weights sequence from the final layer. The monotonic decreasing loss is:

$$\mathcal{L}_{\text{mono}} = \frac{1}{C-1} \sum_{c=2}^{C} \text{ReLU}(\pi_{k,c}^{(L)} - \pi_{k,c-1}^{(L)} + \epsilon), \tag{4}$$

where $\epsilon > 0$ is a small tolerance. The ReLU function penalizes only violations of the expected trend. For an increasing trend, the terms in the parentheses are swapped. In practice, we constrain two prototypes with increasing and decreasing monotonic penalties.

**Non-monotonic Dynamics.** While monotonic constraints are simple and effective for some prototypes, the dynamics of active infections are complex and non-monotonic. Therefore, simply applying a specific predefined pattern to such prototypes can be ineffective. Instead, we regulate their behavior using one of the most fundamental principles in epidemiology: the **basic reproduction number,** $R_0$, which quantifies a pathogen's intrinsic transmissibility. Our goal is to ensure that the infectious dynamics learned by our model correspond to a plausible $R_0$ for the disease being modeled. To achieve this, we introduce a method to compute a differentiable proxy for $R_0$ directly from our model's learned representations. We adapt the classic Next Generation Matrix (NGM) method (Diekmann et al., 2010), denoted as $\mathbf{G}$, based on the Disease-Free Equilibrium time series input. Then, the $R_0$ is defined as the spectral radius of $\mathbf{G}$: $\hat{R}_0^{\text{raw}} = \max_j \|\lambda_j(\mathbf{G})\|$, which has the following lower- and upper-bounds (see proof in Appendix A.4.2):

$$\frac{\sigma_{\min}(\mathbf{F})}{\sigma_{\max}(\mathbf{V})} \le \max_j \|\lambda_j(\mathbf{G})\| = \max_j \|\lambda_j(\mathbf{F}\mathbf{V}^{-1})\| \le \frac{\sigma_{\max}(\mathbf{F})}{\sigma_{\min}(\mathbf{V})}, \tag{5}$$

where $\mathbf{F}$ is the Jacobian of the rates of flows from uninfected to infected classes evaluated at the disease-free equilibrium, and $\mathbf{V}$ is the Jacobian of the rates of all other flows to and from infected compartments, $\lambda_j(\mathbf{G})$ are the eigenvalues of $\mathbf{G}$ and $\sigma_{\min}(\mathbf{V})$ is the smallest singular value of $\mathbf{V}$. Since computing the inverse of matrix $\mathbf{V}$ is not always numerically stable, we approximate the lower- and upper-bound of spectral radius via the singular value ratios $\frac{\sigma_{\min}(\mathbf{F})}{\sigma_{\max}(\mathbf{V})}$ and $\frac{\sigma_{\max}(\mathbf{F})}{\sigma_{\min}(\mathbf{V})}$. Further details are provided in Appendix A.4.1, and here we provide a pseudo code for calculating $R_0$ in Table 1.

---

**Algorithm 1** NGM-PROXY($R_0$): Differentiable $R_0$ bounds

---

**Require:** Encoder $f_{\text{enc}}$, prototypes $\mathbf{E}$, estimator $g$, mix operator $\phi$, disease $d$, range $[R_0^{\text{lo}}, R_0^{\text{hi}}]$.
**Ensure:** Estimates $(\hat{R}_0^{\text{lo}}, \hat{R}_0^{\text{hi}})$, loss $\mathcal{L}_{R_0}$.
  1: **DFE:** Compute DFE embedding $\mathbf{E}_{\text{DFE}} \leftarrow f_{\text{enc}}(\mathbf{X}_{\text{DFE}})$ and weights $\boldsymbol{\pi}^* \leftarrow \text{softmax}(\mathbf{E}_{\text{DFE}}\mathbf{E}^\top)$.
  2: **F:** For each $j = 1, \ldots, K$, compute column $\mathbf{F}_{:,j} \leftarrow \max\{0, g(\phi(\hat{\boldsymbol{\pi}}^{(j)}, \mathbf{E}_{\text{DFE}})) - \boldsymbol{\pi}^*\}$, where $\hat{\boldsymbol{\pi}}^{(j)}$ is a small perturbation on $\boldsymbol{\pi}_j^*$.
  3: **V:** For each $j = 1, \ldots, K$, with $\boldsymbol{\pi}^{\text{evolved}} \leftarrow g(\phi(\boldsymbol{e}_j, \mathbf{E}_{\text{DFE}}))$, compute column $\mathbf{V}_{:,j}$ where $V_{ij} = \max\{0, \pi_i^{\text{evolved}}\}$ for $i \ne j$ and $V_{jj} = 1 - \pi_j^{\text{evolved}}$.
  4: Calculate proxy bounds: $\hat{R}_0^{\text{lo}} \leftarrow \frac{\sigma_{\min}(\mathbf{F})}{\sigma_{\max}(\mathbf{V})}$, $\hat{R}_0^{\text{hi}} \leftarrow \frac{\sigma_{\max}(\mathbf{F})}{\sigma_{\min}(\mathbf{V})}$.
  5: Calibrate estimates $(\hat{R}_0^{\text{lo}}, \hat{R}_0^{\text{hi}}) \leftarrow \text{Calib}_\theta(\hat{R}_0^{\text{lo}}, \hat{R}_0^{\text{hi}})$ and compute the loss:

$$\mathcal{L}_{R_0} \leftarrow \max\{0, R_0^{\text{lo}}(d) - \hat{R}_0^{\text{hi}}\} + \max\{0, \hat{R}_0^{\text{lo}} - R_0^{\text{hi}}(d)\}$$

  6: **return** $(\hat{R}_0^{\text{lo}}, \hat{R}_0^{\text{hi}}, \mathcal{L}_{R_0})$.

---

**Compartment Alignment Loss.** Finally, we combine the above losses into a single alignment objective to enforce epidemiologically meaningful behavior:

$$\mathcal{L}_{\text{align}} = \mathcal{L}_{R_0} + \mathcal{L}_{\text{mono}} + \mathcal{L}_{\text{smooth}}, \quad \mathcal{L}_{\text{smooth}} = \sum_{k=1}^{K} \sum_{c=1}^{C-2} (\pi_{k,c+2}^{(L)} - 2\pi_{k,c+1}^{(L)} + \pi_{k,c}^{(L)})^2, \tag{6}$$

where $\mathcal{L}_{\text{smooth}}$ is a smoothness regularizer that encourages gradual transitions over time.

Table 1: Univariate forecasting results with horizons ranging from 1 to 16 future steps. The lookback window length is set to 36. All models are evaluated over 25 runs, and we report the average MSE and MAE. For CAPE, we also report the 95% confidence interval.

| Dataset | Horizon | LSTM | | GRU | | Dlinear | | Informer | | Autoformer | | MOMENT | | PEM | | PatchTST | | CAPE | |
|---|---|---|---|---|---|---|---|---|---|---|---|---|---|---|---|---|---|---|---|
| | | | | | | | | Transformer-Based (w/ or w/o pre-train) | | | | | | | | | | | |
| | | MSE | MAE | MSE | MAE | MSE | MAE | MSE | MAE | MSE | MAE | MSE | MAE | MSE | MAE | MSE | MAE | MSE | MAE |
| Covid | 1 | 32.290 | 6.749 | **25.222** | 5.739 | 25.799 | 5.476 | 28.412 | 5.778 | 40.465 | 6.837 | 32.026 | 5.029 | 36.163 | 10.136 | 26.824 | 5.159 | 25.841 (±0.129) | **3.611** (±0.011) |
| | 2 | 37.001 | 7.449 | 30.856 | 7.165 | 27.023 | 5.873 | 28.550 | 4.773 | 42.969 | 13.338 | 31.661 | 4.123 | 29.278 | 6.121 | 27.306 | 5.652 | **25.413** (±0.076) | **3.763** (±0.008) |
| | 4 | 38.129 | 8.449 | 29.928 | 8.065 | 30.143 | 6.407 | 45.663 | 8.595 | 46.000 | 7.960 | 35.21 | 5.319 | 33.545 | 8.002 | 25.756 | 5.139 | **24.631** (±0.049) | **3.749** (±0.034) |
| | 8 | 45.500 | 10.680 | 45.337 | 10.595 | 37.733 | 7.393 | 55.651 | 9.945 | 37.424 | 9.758 | 39.633 | 5.940 | 39.577 | 7.547 | 38.908 | 8.998 | **33.003** (±0.033) | **4.827** (±0.005) |
| | 16 | 64.599 | 12.553 | 66.860 | 13.476 | 55.767 | 9.259 | 62.572 | 14.306 | 102.196 | 9.165 | 51.948 | 8.116 | 49.299 | 10.175 | **47.110** | 7.704 | 49.838 (±0.050) | **7.144** (±0.007) |
| | Avg | 43.504 | 9.176 | 39.640 | 9.008 | 35.293 | 6.882 | 44.170 | 8.679 | 53.811 | 9.411 | 38.096 | 5.705 | 37.573 | 8.396 | 33.181 | 6.530 | **31.745** (±0.063) | **4.619** (±0.004) |
| ILI USA | 1 | 0.196 | 0.130 | 0.221 | 0.138 | 0.177 | **0.123** | 0.898 | 0.394 | 0.805 | 0.391 | 0.310 | 0.169 | 0.303 | 0.210 | 0.332 | 0.216 | **0.174** (±0.003) | 0.139 (±0.001) |
| | 2 | 0.281 | 0.156 | 0.322 | 0.167 | 0.224 | 0.148 | 0.395 | 0.229 | 0.806 | 0.399 | 0.328 | 0.176 | 0.328 | 0.193 | 0.283 | 0.180 | **0.192** (±0.002) | **0.141** (±0.001) |
| | 4 | 0.444 | 0.197 | 0.588 | 0.211 | 0.305 | 0.183 | 0.909 | 0.400 | 0.868 | 0.403 | 0.434 | 0.211 | 0.507 | 0.243 | 0.431 | 0.257 | **0.299** (±0.001) | **0.171** (±0.000) |
| | 8 | 0.549 | 0.225 | 0.771 | 0.258 | **0.469** | 0.234 | 0.929 | 0.426 | 0.899 | 0.427 | 0.511 | 0.222 | 0.519 | 0.271 | 0.497 | 0.265 | **0.469** (±0.001) | **0.221** (±0.000) |
| | 16 | 1.515 | 0.332 | 0.946 | 0.287 | **0.595** | 0.269 | 0.690 | 0.351 | 0.970 | 0.410 | 0.709 | **0.259** | 0.682 | 0.324 | 0.651 | 0.311 | 0.650 (±0.001) | 0.278 (±0.000) |
| | Avg | 0.597 | 0.208 | 0.569 | 0.212 | **0.354** | 0.191 | 0.764 | 0.360 | 0.870 | 0.406 | 0.470 | 0.207 | 0.468 | 0.248 | 0.439 | 0.246 | 0.357 (±0.001) | **0.190** (±0.000) |
| ILI Japan | 1 | 0.514 | **0.844** | 0.552 | 0.910 | 0.416 | 0.880 | 2.353 | 1.682 | 0.715 | 1.340 | 0.614 | 3.459 | 0.734 | 1.599 | 0.936 | 1.868 | **0.328** (±0.001) | 0.982 (±0.002) |
| | 2 | 0.758 | 1.384 | 0.745 | **0.904** | **0.648** | 0.956 | 2.446 | 1.767 | 0.928 | 2.786 | 1.490 | 4.111 | 0.919 | 2.214 | 1.531 | 3.443 | 0.709 (±0.001) | 1.206 (±0.002) |
| | 4 | 1.278 | 2.733 | 1.736 | 3.169 | 1.253 | 1.919 | 2.632 | **1.884** | 1.464 | 4.003 | 1.542 | 3.811 | 1.310 | 2.769 | 1.834 | 3.355 | **1.191** (±0.004) | 2.029 (±0.001) |
| | 8 | 1.932 | 1.660 | 1.948 | 2.240 | 1.988 | 2.196 | 2.840 | 1.741 | 1.925 | 1.375 | 2.101 | 2.314 | 1.836 | 1.534 | 2.128 | 1.910 | **1.792** (±0.002) | **1.088** (±0.013) |
| | 16 | 2.118 | 1.657 | 2.097 | 1.550 | 1.884 | 1.517 | 2.490 | 1.633 | 2.438 | 1.799 | 2.314 | 1.255 | 1.936 | 1.315 | 2.265 | 1.698 | **1.878** (±0.002) | **1.163** (±0.001) |
| | Avg | 1.320 | 1.656 | 1.415 | 1.755 | 1.238 | 1.493 | 2.552 | 1.741 | 1.494 | 2.260 | 1.556 | 2.99 | 1.347 | 1.886 | 1.739 | 2.455 | **1.179** (±0.002) | **1.294** (±0.004) |
| Measles | 1 | 0.191 | 1.076 | 0.202 | 1.249 | 0.207 | 1.022 | 0.428 | 2.188 | 0.699 | 2.733 | 0.207 | 1.270 | 0.330 | 1.366 | 0.257 | 1.297 | **0.111** (±0.005) | **0.615** (±0.002) |
| | 2 | 0.230 | 1.249 | 0.251 | 1.147 | 0.232 | 1.183 | 0.479 | 2.094 | 0.584 | 1.672 | 0.248 | 1.359 | 0.350 | 1.699 | 0.418 | 1.509 | **0.157** (±0.003) | **1.078** (±0.002) |
| | 4 | 0.261 | **1.153** | 0.304 | 1.175 | 0.297 | 1.442 | 1.639 | 3.510 | 0.851 | 2.005 | 0.296 | 1.395 | 0.464 | 2.011 | 0.459 | 1.512 | **0.188** (±0.003) | 1.352 (±0.001) |
| | 8 | 0.415 | 2.007 | **0.392** | **1.703** | 0.468 | 1.938 | 0.592 | 2.627 | 1.171 | 2.767 | 0.476 | 1.771 | 0.726 | 2.587 | 0.721 | 2.558 | 0.406 (±0.002) | 2.002 (±0.000) |
| | 16 | **0.696** | **2.431** | 0.729 | 2.695 | 0.953 | 2.864 | 2.098 | 3.793 | 1.922 | 3.924 | 0.763 | 2.747 | 1.213 | 3.228 | 1.271 | 3.164 | 0.883 (±0.001) | 2.836 (±0.000) |
| | Avg | 0.358 | 1.583 | 0.375 | 1.594 | 0.431 | 1.690 | 1.047 | 2.843 | 1.046 | 2.620 | 0.398 | 1.708 | 0.616 | 2.178 | 0.625 | 2.008 | **0.349** (±0.003) | **1.576** (±0.002) |
| Dengue | 1 | 0.583 | 1.579 | 0.627 | 1.343 | 0.503 | **1.074** | 0.627 | 1.697 | 1.556 | 2.456 | 0.630 | 1.435 | 0.912 | 1.792 | 2.056 | 2.600 | **0.367** (±0.011) | 1.282 (±0.003) |
| | 2 | 0.634 | 1.417 | 0.676 | 1.604 | 0.566 | 1.337 | 0.905 | 2.111 | 1.827 | 2.622 | 0.690 | 1.597 | 0.844 | 1.709 | 0.925 | 1.567 | **0.317** (±0.003) | **1.272** (±0.006) |
| | 4 | 0.823 | 2.137 | 0.984 | 1.940 | 0.845 | 1.767 | 1.170 | 2.184 | 2.546 | 2.887 | 0.938 | 1.827 | 1.236 | 2.270 | 1.419 | 2.226 | **0.508** (±0.003) | **1.534** (±0.003) |
| | 8 | 1.534 | 2.758 | 1.375 | 2.623 | 1.488 | 2.410 | 1.392 | 2.428 | 3.322 | 3.322 | 1.504 | 2.283 | 1.806 | 2.587 | 1.581 | 2.261 | **1.169** (±0.004) | **2.104** (±0.002) |
| | 16 | 2.561 | 3.078 | 2.745 | 2.914 | 2.861 | 3.003 | 3.841 | 3.454 | 4.734 | 3.581 | 2.768 | 2.932 | 2.938 | 3.116 | 4.923 | 3.618 | **2.512** (±0.002) | **2.784** (±0.006) |
| | Avg | 1.227 | 2.194 | 1.281 | 2.085 | 1.252 | 1.918 | 1.587 | 2.375 | 2.868 | 2.973 | 1.306 | 2.015 | 1.547 | 2.295 | 2.181 | 2.454 | **0.975** (±0.005) | **1.795** (±0.004) |

Table 2: Few-shot learning results with horizons ranging from 1 to 16 future steps. The length of the lookback window is set to 36. Each model is evaluated after being trained on 20%, 40%, 60%, and 80% of the full training data. $\Delta(\%)$ stands for the relative improvement after training with 20% more data in terms of average MSE over all horizons. The full result is shown in Appendix A.11.

| Dataset | CAPE | | | | | PatchTST | | | | | Dlinear | | | | | MOMENT | | | | | PEM | | | | |
|---|---|---|---|---|---|---|---|---|---|---|---|---|---|---|---|---|---|---|---|---|---|---|---|---|---|
| | 20% | 40% | 60% | 80% | 100% | 20% | 40% | 60% | 80% | 100% | 20% | 40% | 60% | 80% | 100% | 20% | 40% | 60% | 80% | 100% | 20% | 40% | 60% | 80% | 100% |
| ILI USA | 2.121 | 1.400 | 0.760 | 0.369 | 0.309 | 2.114 | 1.219 | 0.677 | 0.401 | 0.373 | 2.822 | 1.594 | 0.816 | 0.412 | 0.346 | 3.990 | 1.847 | 0.913 | 0.459 | 0.381 | 2.143 | 1.261 | 0.681 | 0.419 | 0.353 |
| Δ(%) | - | 33.99% | 45.71% | 51.45% | 16.26% | - | 42.34% | 44.45% | 40.77% | 6.98% | - | 43.53% | 48.78% | 49.51% | 16.02% | - | 53.69% | 50.58% | 49.72% | 17.00% | - | 41.13% | 46.00% | 38.33% | 15.76% |
| Dengue | 13.335 | 6.386 | 2.356 | 1.511 | 0.892 | 13.712 | 7.304 | 2.771 | 1.678 | 0.984 | 15.828 | 8.420 | 2.850 | 1.748 | 1.080 | 15.697 | 7.536 | 2.816 | 1.733 | 1.358 | 12.90 | 7.055 | 2.745 | 1.707 | 0.964 |
| Δ(%) | - | 52.07% | 63.12% | 35.87% | 40.95% | - | 46.72% | 62.06% | 39.43% | 41.39% | - | 46.81% | 66.15% | 38.64% | 38.19% | - | 52.00% | 62.63% | 38.45% | 21.65% | - | 45.32% | 61.09% | 37.79% | 43.51% |
| Measles | 0.483 | 0.600 | 0.381 | 0.285 | 0.269 | 0.863 | 0.834 | 0.448 | 0.359 | 0.306 | 1.194 | 1.130 | 0.602 | 0.478 | 0.394 | 1.661 | 0.915 | 0.425 | 0.471 | 0.500 | 0.670 | 0.896 | 0.430 | 0.364 | 0.306 |
| Δ(%) | - | -24.22% | 36.50% | 25.20% | 5.61% | - | 3.36% | 46.25% | 19.91% | 14.81% | - | 5.36% | 46.64% | 20.63% | 17.58% | - | 44.91% | 53.55% | -10.59% | -6.16% | - | -33.87% | 51.91% | 15.35% | 15.93% |

## 3.4 OPTIMIZATION SCHEME

To stably train the model, we employ an alternating optimization strategy. We alternate between freezing the main model to update the prototype embeddings $\mathbf{E}$, and then freezing $\mathbf{E}$ to update the main model parameters. This process is applied across two training phases. During pre-training, we use the self-supervised objectives $\mathcal{L}_{\text{recon}}$ and $\mathcal{L}_{\text{CL}}$, as well as the epidemic-aware regularization $\mathcal{L}_{\text{align}}$, which gives $\mathcal{L}_{\text{pretrain}} = \mathcal{L}_{\text{recon}} + \mathcal{L}_{\text{CL}} + \lambda\mathcal{L}_{\text{align}}$, where $\lambda$ is a hyperparameter. During finetuning on a specific type of pathogen, we adopt the forecasting loss (MSE) in place of the self-supervised objective, giving $\mathcal{L}_{finetune} = \text{MSE}(\mathbf{y} - \hat{\mathbf{y}}) + \lambda\mathcal{L}_{\text{align}}$.

## 4 EXPERIMENT

### 4.1 EXPERIMENT SETUP

**Datasets.** For pre-training our model and PatchTST, we manually collected 17 distinct weekly-sampled diseases from Project Tycho (van Panhuis et al., 2018). For evaluation, we utilize five downstream datasets covering various diseases and locations: ILI USA (Centers for Disease Control and Prevention, 2023a), ILI Japan (National Institute of Infectious Diseases, 2023), COVID-19 USA (Dong et al., 2020), Measles England (Lau et al., 2020), and Dengue across countries (Open-Dengue, 2023). Additionally, RSV (Centers for Disease Control and Prevention, 2023c) and MPox (Centers for Disease Control and Prevention, 2023b) infections in the US are used to test zero-shot performance. More details can be found in Appendix A.5.

**Preprocessing.** For pre-training datasets, we aggregate the time series based on time and locations to acquire the national-level infection trajectory. For all the datasets used in this study, we examine the infection trajectory for all diseases and locations, and filter the time series with extremely short observations or a large number of missing values to form a high-quality evaluation testbed. Then, we split the datasets into train/val/test sets and perform normalization on the time series.

**Baselines** We adopt baseline models from the comprehensive *EpiLearn* toolkit (Liu et al., 2024a), comparing our model against two categories: *non-pretrained* and *pre-trained* models. *Non-pretrained* baselines including RNN-based approaches like LSTM and GRU (Wang et al., 2020; Natarajan et al., 2023), the MLP-based model DLinear (Zeng et al., 2023), and transformer-based architectures (Wu et al., 2021; Zhou et al., 2021; 2022). *Pre-trained* baselines include state-of-the-art models such as PatchTST (Nie et al., 2022) and the time series foundation model MOMENT (Goswami et al., 2024). We provide further comparisons with PEM (Kamarthi & Prakash, 2023) in the few-shot setting and ARIMA (Panagopoulos et al., 2021) in the online forecasting setting in Appendix A.8. We perform hyperparameter tuning on the learning rate and weight decay for all models. ( See A.6 for details)

**Research Questions.** In the following experiment, we propose and answer the following questions: *Q1:* How does the latent compartment contribute to epidemic forecasting? *Q2:* How does the model perform on diverse downstream datasets compared to other general time series models? *Q3:* How does the model perform in the few-shot setting with fewer observations? *Q4:* Does the proposed epidemic-aware regularization help? *Q5:* How does pre-training influence downstream performance?

## 4.2 SIMULATION ON MECHANISTIC MODEL

Before conducting the empirical experiment, we first validate that CAPE's latent compartments behave as intended under controlled simulations, which answers **Q1**. We constructed a simulation dataset based on the Susceptible-Infectious-Recovered-Deceased (SIRD) model for analysis. In this scenario, S, R, and D represent compartments with monotonic increasing or decreasing behaviors, while I shows a single peak pattern. We initialize our model with 16 latent compartments and assign S, I, R each with three compartment prototypes, D with two prototypes, and the rest without constraints. As shown in Figure 2, we observe that the trend and magnitude of latent compartment contribution roughly align with the actual compartments, which verifies the usefulness of our model.

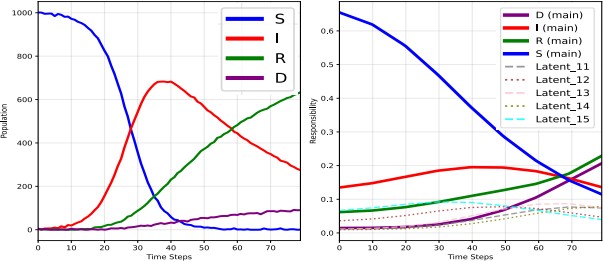
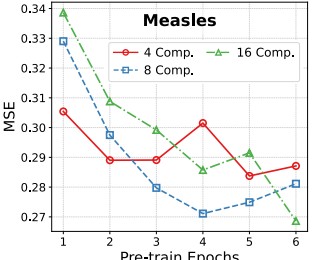

Figure 2: Simulation on SIRD model. Left: Ground-truth trajectory; Right: Inferred compartment contributions.

Figure 3: Downstream performance vs. compartments distribution(See A.12).

## 4.3 FINE-TUNING (FULL-SHOT SETTING)

To answer **Q2**, we finetune our model on diverse downstream datasets and compare performance across baselines from various designs. For non-pre-trained models, we train the entire model on the training split, while for pre-trained models, we fine-tune a few epochs on downstream datasets by transferring the task-specific head $h_\psi$ from pre-training to the forecasting task. We evaluate short-term and long-term performance by reporting mean MSE and MAE across horizons from 1 to 16 under 25 runs on test evaluation. From Table 1, we observe that CAPE achieves the best average performance across all downstream datasets compared with baselines.

## 4.4 DATA SCARCE SCENARIO, AND ABLATION STUDY

To answer **Q3** and **Q4**, we evaluate the models in the few-shot and zero-shot regime with limited or no fine-tuning data, and perform ablations isolating the effects of pre-training and the epidemic-aware regularization.

Table 3: Zero-shot performance with a lookback window length of 12. All results are averaged over 4 weeks or days in the future. $\Delta(\%)$ stands for the relative improvement over the baselines.

| Dataset | $\Delta$ (%) | CAPE | PatchTST | MOMENT |
|---|---|---|---|---|
| ILI USA | 9.26% | **0.147** | **0.164** | 0.549 |
| ILI Japan | 17.06% | **0.705** | **0.907** | 2.062 |
| Measles | 3.97% | **0.145** | **0.167** | 0.533 |
| MPox | 20.00% | **0.0004** | **0.0005** | 0.0013 |
| Dengue (mixed) | 10.17% | **0.371** | **0.427** | 1.624 |
| RSV | 26.06% | **0.834** | **1.128** | 1.849 |
| Covid (daily interval) | 13.80% | **5.173** | **6.001** | 18.881 |

Table 4: Ablation study on removing pre-training and epidemic-aware regularization. Results are averaged over 25 runs of evaluation.

| Dataset | H | w/o Pre-train | | w/o Epidemic Reg. | | CAPE | |
|---|---|---|---|---|---|---|---|
| | | MSE | MAE | MSE | MAE | MSE | MAE |
| ILI USA | 1 | 0.148 | 0.114 | 0.180 | 0.144 | 0.174 | 0.139 |
| | 2 | 0.229 | 0.151 | 0.200 | 0.145 | 0.192 | 0.141 |
| | 4 | 0.409 | 0.202 | 0.297 | 0.171 | 0.299 | 0.171 |
| | 8 | 0.575 | 0.241 | 0.565 | 0.240 | 0.469 | 0.221 |
| | 16 | 0.640 | 0.289 | 0.652 | 0.278 | 0.650 | 0.278 |
| | Avg | 0.400 | 0.199 | 0.379 | 0.196 | **0.357** | **0.190** |
| ILI Japan | 1 | 0.371 | 0.406 | 0.334 | 1.130 | 0.328 | 0.982 |
| | 2 | 0.677 | 1.141 | 0.703 | 1.235 | 0.709 | 1.206 |
| | 4 | 1.300 | 1.540 | 1.284 | 1.537 | 1.191 | 2.029 |
| | 8 | 1.835 | 1.190 | 1.798 | 1.082 | 1.792 | 1.088 |
| | 16 | 1.920 | 1.232 | 1.866 | 1.149 | 1.878 | 1.163 |
| | Avg | 1.221 | 1.102 | 1.197 | 1.227 | **1.179** | 1.294 |

**Few-Shot Forecasting.** Predicting outbreaks of new diseases or in unfamiliar locations is difficult for purely data-driven models with limited data, making few-shot and zero-shot forecasting essential. To simulate this, we reduce training data to [20%, 40%, 60%, 80%] and report average MSE over 1–16 time steps (Table 2). Key observations: (a) More training data consistently improves performance. (b) CAPE achieves the best results in most cases, showing strong few-shot capability. (c) Dlinear underperforms at 20% data compared to epidemic-pretrained models, but surpasses MOMENT on ILI USA and Measles when both are trained on 20%, highlighting the value of pre-training on epidemic time series. (More details are shown in A.11).

**Zero-Shot Forecasting.** We evaluate CAPE in a zero-shot setting by freezing transformer-based models with their pre-training heads. All models receive a 12-step historical input and predict the next 4 steps (Table 3). Key observations: (a) CAPE consistently outperforms baselines, confirming superior zero-shot ability. (b) Epidemic-specific pre-training yields better results than general pre-training (e.g., MOMENT), underscoring the importance of domain-specific data.

**Ablation Study.** We evaluate the contributions of CAPE's components in Table 4. Removing pre-training leads to the largest degradation, with average MSE increasing from 0.357 to 0.400 on ILI USA. Also, dropping epidemic regularization hurts performance across most horizons and the average MSE (e.g., 1.179 to 1.197 on ILI Japan). Overall, CAPE consistently achieves the lowest errors, showing that both pre-training and epidemic regularization are important, with pre-training providing the greater benefit.

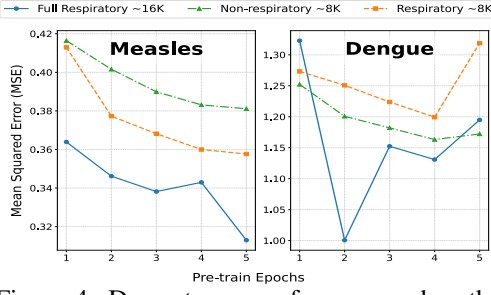

Figure 4: Downstream performance when the model is pre-trained with either respiratory or non-respiratory data.

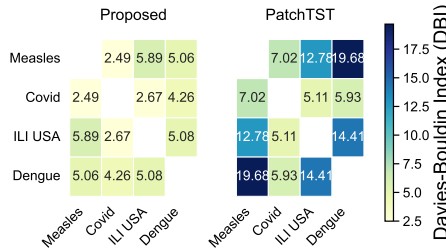

Figure 5: DBI between the embeddings of each pair of downstream datasets from the pre-trained model. (See Figure 7 for visualization.).

## 4.5 DEEPER ANALYSIS

To answer **Q5**, we (i) examine the representation quality, transferability, and robustness against distribution shift, and (ii) further explore the power of pre-training from two perspectives: compute budget and pre-training materials.

**Transferability. (a) Cross-Location:** While we pre-train our model with influenza data from the USA, the few-shot and zero-shot evaluation on the influenza outbreak in Japan also shows superior performance, underscoring the crucial role of pre-training in enabling generalization to novel regions. **(b) Cross-Disease:** While we include various types of diseases in our pre-training dataset, novel

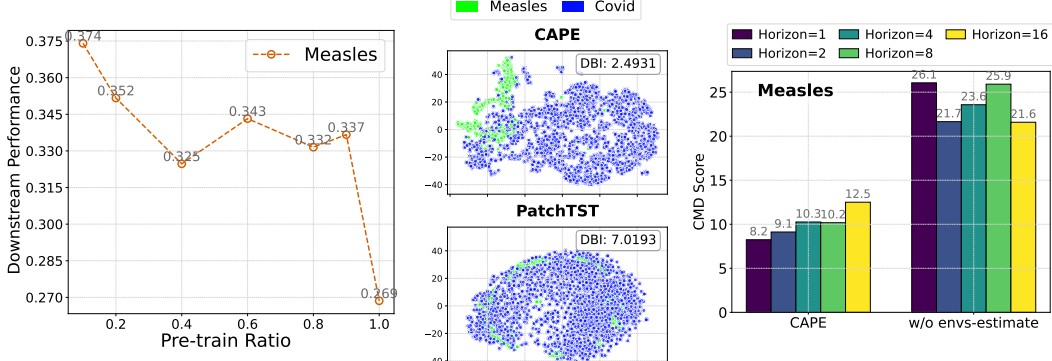

Figure 6: Downstream performance across pre-training ratios. More datasets are in A.13.

Figure 7: Representation learned by CAPE vs. PatchTST after pre-training.

Figure 8: CMD scores w/ and w/o compartment estimate between train/test sets. (A.14)

diseases, like COVID-19, that are unseen in the pre-training stage, are incorporated during the downstream evaluation. The ability of our model to adapt to novel diseases is proven compared to the version not pre-trained on the COVID dataset (Table 3), which surpasses the MOMENT that is not pre-trained on other diseases by 72.60%.

**Representation Learning.** To evaluate the representation quality learned during pre-training, we compute the DBI score of sample representations from different diseases. As shown in Figure 5 and 7, compared to PatchTST, CAPE is able to distinguish the representations from different diseases, effectively capturing the diverse underlying dynamics of different pathogens.

**Tackling Distribution Shift.** We define distribution shifts as changes in infection patterns from training to test data. We compute the Central Moment Discrepancy (CMD)(Zellinger et al., 2017) between training and test distributions for each disease. As shown in Figure 8, compared to the version without compartmental prototypes, CAPE achieves the lowest CMD scores, highlighting its effectiveness in mitigating distribution shifts.

**Analysis on Pre-training. (a) Impact of Compute During Pre-training.** Evaluating four downstream datasets (Figure 3), we find that increasing pre-training epochs consistently improves performance on the Measles dataset. Additionally, models with more compartment prototypes $K$ perform better as pre-training epochs increase. **(b) Impact of Pre-Training Materials.** We examine potential biases in our pre-training dataset by splitting it into respiratory and non-respiratory diseases. As shown in Figure 4, with similar volumes of pre-training data, the model performs better when the tested disease types align with the pre-training data. However, the size of the pre-training material has a stronger impact. **(c) Impact of Pre-Training Material Scale:** To explore how the pre-training material scale affects downstream performance, we scaled the original pre-training dataset and tested it on downstream datasets. As shown in Figure 6, a sudden performance boost is observed at around a 60% reduction for both Measles and Dengue datasets.

## 5 CONCLUSION

We present CAPE, the first open-source pre-training framework for epidemic forecasting that learns flexible latent population states, termed compartmental prototypes, to address structural heterogeneity, hidden population states, and distribution shifts in epidemic pre-training. By designing large-scale epidemic self-supervised objectives with lightweight epidemic-aware regularization, CAPE captures transferable dynamics across diseases and regions. Extensive experiments demonstrate state-of-the-art generalization in zero-shot, few-shot, and cross-disease settings. In the future, we plan to extend CAPE to incorporate spatial dynamics for richer generalization, integrate principled approaches to uncertainty quantification, and further enhance the interpretability of the pre-trained model, ultimately advancing toward trustworthy and actionable epidemic forecasting.

ETHICS STATEMENT

**We have adhered to the ICLR Code of Ethics in preparing this submission.** This work does not involve human subjects, personally identifiable data, or sensitive information. All datasets used are publicly available benchmark datasets, and we follow their respective usage and licensing guidelines. The proposed methods are designed for advancing research in high-dimensional time series forecasting and do not raise foreseeable risks of harm.

REPRODUCIBILITY STATEMENT

We provide an anonymous repository containing the full source code and implementation details of our proposed CAPE at `https://anonymous.4open.science/r/CAPE_ICLR26-A041/`. Detailed descriptions of model architectures, training protocols, and hyperparameters are included in the main text and appendix. These resources are intended to ensure that all reported results can be independently reproduced.

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

# A  APPENDIX

## A.1  POTENTIAL IMPACT

This paper advances the interdisciplinary fields of machine learning and epidemiology by enhancing the accuracy of epidemic forecasting in data-limited settings. We outline the potential impacts as follows:

**Early Insights.** We provide novel insights into how pre-training and compartment modeling improve epidemic forecasting. Our results demonstrate that pre-training significantly enhances model accuracy, with gains increasing as more pre-training data is incorporated. This finding paves the way for future research to develop foundational models in epidemic analysis using larger datasets. Additionally, we confirm the importance of accounting for both inherent disease dynamics and compartmental factors to achieve robust forecasting performance.

**Social Impact.** Epidemic time series data are often sparse due to limited sampling rates, hindering public health organizations' ability to accurately predict infections during novel disease outbreaks. This paper addresses this challenge by showcasing the few-shot and zero-shot forecasting capabilities of pre-trained models. These capabilities can provide powerful tools for early warning and timely intervention, ultimately supporting more effective public health responses and safeguarding communities against emerging infectious diseases.

## A.2  LIMITATIONS

While CAPE shows strong performance, it has several limitations. First, it does not currently incorporate uncertainty estimation, which is important for risk-aware decision-making. Second, its effectiveness may be constrained by the scale and diversity of available data. Lastly, the learned compartment representations lack interpretability, limiting their transparency and potential for insight in public health contexts.

## A.3  USE OF LLMS

We use LLMs to check grammar and polish the language of this paper for clarity.

## A.4  THEORETICAL ANALYSIS

### A.4.1  INFERENCE OF $R_0$

**(1) Disease-Free Equilibrium (DFE):** Calculation of $\mathbf{F}$ and $\mathbf{V}$ requires defining the population flow, and since our framework does not utilize explicit population flows, we use the learned latent compartment contributions, $\boldsymbol{\pi}$, as a potential *differentiable proxy that correlates with population*, as demonstrated in Figure 2. Specifically, we establish the compartment contributions at DFE: $\boldsymbol{\pi}^*$, which represents a scenario with no ongoing epidemic, by feeding a zero-infection time series, $\mathbf{X}_{\text{DFE}} \in \mathbb{R}^{T \times 1}$, into the encoder:

$$\boldsymbol{\pi}^* = \frac{1}{C} \sum_{c=1}^{C} \text{softmax}(\boldsymbol{E}_{\text{DFE}} \mathbf{E}^T[c,:]) \in \mathbb{R}^K, \quad \boldsymbol{E}_{\text{DFE}} \leftarrow f_{\text{enc}}(\mathbf{X}_{\text{DFE}}), \tag{7}$$

where $\boldsymbol{E}_{\text{DFE}}$ are the weighted sum of latent compartment prototypes at the DFE.

**(2) Calculation of F:** The $\mathbf{F}$ matrix, which quantifies new infections, is estimated by applying perturbation on each infectious compartment: $\mathbf{H}_{\text{pert},(j)} = \phi(\hat{\boldsymbol{\pi}}, \mathbf{E}_{\text{DFE}}) = (1 - \alpha)\mathbf{E}_{\text{DFE}} + \alpha \hat{\boldsymbol{\pi}}^T \mathbf{E}$, where $\alpha = 0.1$, $\hat{\boldsymbol{\pi}} = \frac{\boldsymbol{\pi}^* + \boldsymbol{\epsilon}_j}{\|\boldsymbol{\pi}^* + \boldsymbol{\epsilon}_j\|_1}$ and $\boldsymbol{\epsilon}_j$ is a perturbation applied on the entry (compartment) $j$. Therefore, we get the element of $\mathbf{F}$ via:

$$F_{ij} = \max(0, \pi_i^{\text{new},(j)} - \pi_i^*), \quad \boldsymbol{\pi}^{\text{new},(j)} = g(\mathbf{H}_{\text{pert},(j)}). \tag{8}$$

**(3) Calculation of V:** $\mathbf{V}$ characterizes transition rates out of infectious compartments, where $V_{jj}$ represents the total departure rate from compartment $j$ and $V_{ij}$ captures transitions from compartment $j$ to compartment $i$. For each infectious compartment $j$, we initialize with unit mass $\boldsymbol{\pi}^{\text{unit, (j)}}$ and

acquire the updated contributions in a similar way: $\boldsymbol{\pi}^{\text{evolved},(j)} = g(\phi(\boldsymbol{\pi}^{\text{unit, (j)}}, \mathbf{E}_{\text{DFE}}))$. Then the V matrix elements are computed as:

$$V_{ij} = \begin{cases} \max(0, \pi_i^{\text{evolved},(j)}) & \text{if } i \neq j, \\ \pi_j^{\text{unit},(j)} - \pi_j^{\text{evolved},(j)} & \text{if } i = j, \end{cases} \tag{9}$$

where $V_{ij}$ measures mass appearing in compartment $i$ when $j$ initially contains unit mass, and $V_{jj}$ quantifies the total rate of departure from compartment $j$. Finally, a linear layer is applied to align the scale of the estimated lower- and upper-bound with the ground-truth range of $R_0$ and the loss is computed as: $\mathcal{L}_{R_0} = \text{RELU}(\hat{R}_0^{\text{lower}}, R_0^{\text{lower}}) + \text{RELU}(\hat{R}_0^{\text{upper}}, R_0^{\text{upper}})$. This NGM-based approach provides a theoretically grounded method for computing $R_0$ that respects the compartmental structure while being differentiable for end-to-end training.

### A.4.2 PROOF OF EQUATION 5

**Setup and Preliminaries.** Let $V \in \mathbb{C}^{m \times n}$. The (operator) 2-norm of a matrix $M$ is

$$\|M\|_2 = \sup_{x \neq 0} \frac{\|Mx\|_2}{\|x\|_2} = \max_{\|x\|_2 = 1} \|Mx\|_2.$$

The *singular values* $\sigma_1(V) \geq \cdots \geq \sigma_r(V) \geq 0$ (with $r = \text{rank}(V)$) are, by definition, the nonnegative square roots of the eigenvalues of $V^*V$ (where $*$ denotes conjugate transpose), counted with multiplicity and ordered nonincreasingly.

For any matrix $M$ and any vector $x$,

$$\sigma_{\min}(M) \|x\|_2 \leq \|Mx\|_2 \leq \sigma_{\max}(M) \|x\|_2, \quad \|M\|_2 = \sigma_{\max}(M).$$

For square $M$, the singular values $\{\sigma_i(M)\}_{i=1}^n$ and eigenvalues $\{\lambda_i(M)\}_{i=1}^n$ satisfy

$$\prod_{i=1}^n \sigma_i(M) = \sqrt{\det(M^*M)} = |\det M| = \prod_{i=1}^n |\lambda_i(M)|.$$

In particular, by comparing geometric means to extrema,

$$\sigma_{\min}(M) \leq \Big( \prod_{i=1}^n \sigma_i(M) \Big)^{1/n} = \Big( \prod_{i=1}^n |\lambda_i(M)| \Big)^{1/n} \leq \max_i |\lambda_i(M)| = \rho(M). \tag{P1}$$

Thus, for any square $M$,

$$\sigma_{\min}(M) \leq \rho(M) \leq \sigma_{\max}(M) = \|M\|_2. \tag{P2}$$

**Theorem A.1** (Bounds for $R_0$). *Let $F, V \in \mathbb{C}^{n \times n}$ with $V$ invertible, and define*

$$R_0 \equiv \rho(FV^{-1}),$$

*where $\rho(\cdot)$ denotes the spectral radius, $\| \cdot \|_2$ the operator 2-norm, and $\sigma_{\max}(\cdot), \sigma_{\min}(\cdot)$ the maximal and minimal singular values, respectively. Then $R_0$ satisfies the bounds*

$$\boxed{\frac{\sigma_{\min}(F)}{\sigma_{\max}(V)} \leq \rho(FV^{-1}) \leq \frac{\sigma_{\max}(F)}{\sigma_{\min}(V)}.}$$

**Derivation for Lower bound.**

*Proof.* For any compatible $A, B$ and any unit vector $x$,

$$\|ABx\|_2 \geq \sigma_{\min}(A) \|Bx\|_2 \geq \sigma_{\min}(A) \sigma_{\min}(B) \|x\|_2,$$

hence

$$\sigma_{\min}(AB) \geq \sigma_{\min}(A) \sigma_{\min}(B). \tag{S1}$$

Apply (S1) with $A = F$ and $B = V^{-1}$ (with $V$ invertible):

$$\sigma_{\min}(FV^{-1}) \geq \sigma_{\min}(F) \sigma_{\min}(V^{-1}).$$

Using the inversion identity for singular values,

$$\sigma_{\min}(V^{-1}) = \frac{1}{\sigma_{\max}(V)},$$

we obtain

$$\boxed{\sigma_{\min}(FV^{-1}) \geq \frac{\sigma_{\min}(F)}{\sigma_{\max}(V)}.} \tag{L1}$$

By (P1) applied to $M = FV^{-1}$, we get:

$$\sigma_{\min}(FV^{-1}) \leq \rho(FV^{-1}). \tag{L2}$$

Combining (L1) and (L2) yields the rigorous lower bound

$$\boxed{\frac{\sigma_{\min}(F)}{\sigma_{\max}(V)} \leq \rho(FV^{-1}).} \tag{LB}$$

$\square$

**Derivation for Upper bound.**

*Proof.* For any square $M$, $\rho(M) \leq \|M\|_2$ by (P2). Thus

$$\rho(FV^{-1}) \leq \|FV^{-1}\|_2 \leq \|F\|_2 \|V^{-1}\|_2 = \sigma_{\max}(F)\frac{1}{\sigma_{\min}(V)}.$$

Hence the rigorous upper bound is

$$\boxed{\rho(FV^{-1}) \leq \frac{\sigma_{\max}(F)}{\sigma_{\min}(V)}.} \tag{UB}$$

**Final result.** Putting (LB) and (UB) together, we obtain

$$\boxed{\frac{\sigma_{\min}(F)}{\sigma_{\max}(V)} \leq \rho(FV^{-1}) \leq \frac{\sigma_{\max}(F)}{\sigma_{\min}(V)}.}$$

$\square$

### A.5 Pre-train and Downstream Datasets Details

Due to incomplete records and limited non-U.S. data, we curated diverse, high-quality datasets emphasizing:

- Epidemic variation: Decades of weekly data (visualization) reflect multi-wave trends and interventions.
- Geographic spread: Covers 50 U.S. states and 25 non-U.S. Dengue regions.
- Climate diversity: Implicit variation (e.g., tropical Florida vs. temperate Maine).
- Data quality: Manually curated, gap-free datasets (from CDC, ProjectTycho, etc.).

We collect the rough range of $R_0$ for each pathogen from various prior research and apply z-score normalization for all datasets.

Table 5: Pre-training datasets from Project Tycho with basic reproduction number ($R_0$) ranges.

| Disease | Number of States | Total Length | Non-Respiratory | $R_0$ Range |
|---|---|---|---|---|
| Gonorrhea | 39 | 37,824 | Yes | 1.00–1.01 |
| Meningococcal Meningitis | 37 | 44,890 | No | 0.6–1.6 |
| Varicella | 30 | 33,298 | No | 10–12 |
| Typhoid Fever | 44 | 89,868 | Yes | 2.8–7.0 |
| Acute Poliomyelitis | 47 | 74,070 | Yes | 5–7 |
| Hepatitis B | 31 | 34,322 | Yes | 1.0–3.3 |
| Pneumonia | 41 | 68,408 | No | 1.4–1.4 |
| Hepatitis A | 38 | 37,303 | Yes | 1.1–3.5 |
| Influenza | 42 | 61,622 | No | 1.2–1.6 |
| Scarlet Fever | 48 | 129,460 | No | 0.6–2.0 |
| Smallpox | 44 | 71,790 | No | 3.5–6 |
| Tuberculosis | 39 | 95,564 | No | 0.24–4.3 |
| Measles | 50 | 151,867 | No | 12–18 |
| Diphtheria | 46 | 112,037 | No | 1.7–4.3 |
| Mumps | 41 | 50,215 | No | 4–7 |
| Pertussis | 46 | 109,761 | No | 12–17 |
| Rubella | 7 | 6,274 | No | 3.4–7.0 |

Table 6: Statistics of the downstream datasets for evaluation.

| Disease | Number of Regions | Sampling Rate | Respiratory | Total Length | $R_0$ |
|---|---|---|---|---|---|
| ILI USA | 1 | Weekly | Yes | 966 | 1.2-1.4 |
| ILI Japan | 1 | Weekly | Yes | 348 | 1.0-2.0 |
| Measles | 1 | Bi-weekly | Yes | 1,108 | 12-18 |
| Dengue | 23 | Mixed | No | 10,739 | 3.12-5.39 |
| RSV | 13 | Weekly | Yes | 4,316 | 1-5 |
| MPox | 1 | Daily | No | 876 | 1.1-2.7 |
| COVID | 16 | Daily | Yes | 12,800 | 2.9-9.5 |

### A.5.1 PRE-TRAIN DATASETS

In this study, we utilize a comprehensive collection of 17 distinct diseases from the United States, sourced from Project Tycho. These diseases encompass both respiratory and non-respiratory categories and serve as the foundation for pre-training two transformer-based models: **CAPE**, and **PatchTST**. The selection criteria for these datasets were meticulously chosen based on the following factors:

**Temporal Coverage and Geographic Representation**: We prioritized diseases with extensive time series data and coverage across multiple regions to ensure the models are trained on diverse and representative datasets.

**Consistent Sampling Rate**: All selected datasets maintain a uniform sampling rate, which is crucial for the effective training of transformer models that rely on temporal patterns.

**Data Quantity**: Diseases with larger datasets in terms of both temporal length and the number of regions were preferred to enhance the robustness and generalizability of the models.

Among the 17 diseases, five are classified as non-respiratory, providing a balanced representation that allows the models to learn from varied disease dynamics. Before the pre-training phase, each disease dataset underwent a normalization process to standardize the data scales, ensuring comparability across different diseases. Subsequently, the datasets were aggregated at the national level based on their corresponding timestamps. The details of the pre-training datasets are summarized in Table 5.

### A.5.2 DOWNSTREAM DATASETS

In addition, we collect seven datasets of different types of diseases from diverse sources for downstream evaluations, which are all normalized without further processing. A summary of the downstream datasets is shown in Table 6.

All collected diseases can be categorized into **Respiratory** and **Non-respiratory** types, which differ in their modes of transmission:

**Respiratory.** Respiratory diseases are transmitted primarily through the air via aerosols or respiratory droplets expelled when an infected individual coughs, sneezes, or talks. These diseases predominantly affect the respiratory system, including the lungs and throat.

**Non-respiratory.** Non-respiratory diseases are transmitted through various other routes such as direct contact, vectors (e.g., mosquitoes, ticks), contaminated food or water, and sexual activities. These diseases can affect multiple body systems and have diverse transmission pathways unrelated to the respiratory system.

A more detailed description of each dataset is shown below:

- **ILI USA (Centers for Disease Control and Prevention, 2023a)**: The weekly influenza-like-illness infection was reported by the CDC in the United States. We use the national-level infection counts from 2002 to 2020, which include various disease such as H1N1, H3N2v, etc.
- **ILI Japan (National Institute of Infectious Diseases, 2023)**: This dataset is collected from the Infectious Diseases Weekly Report (IDWR) in Japan, which contains national counts of weekly influenza-like-illness infections from August 2012 to March 2019.
- **Measles (Lau et al., 2020)**: The measles dataset contains biweekly measles infections in England from 1906 to 1948.
- **Dengue (OpenDengue, 2023)**: OpenDenue aims to build and maintain a database of dengue case counts for every dengue-affected country worldwide since 1990 or earlier. We selected 23 countries for the experiment, which reports daily to weekly infections.
- **RSV (Centers for Disease Control and Prevention, 2023c)**: The Respiratory Syncytial Virus (RSV) infections in the US are reported by the RSV-NET from CDC. We use the weekly infections across 13 states from 2016 to 2024.
- **MPox (Centers for Disease Control and Prevention, 2023b)**: The clade II MPox case trends data in the US is reported by CDC. We use the nationwide weekly infections from 2022 to 2024.
- **COVID (Dong et al., 2020)**: The original data is from the Novel Coronavirus Visual Dashboard operated by the Johns Hopkins University Center for Systems Science and Engineering (JHU CSSE). We use the daily COVID-19 infections collected by JHU from 2020 to 2022 across 16 states.

### A.6    IMPLEMENTATION DETAILS

**Motivation for Our Settings.** Our primary focus is on pre-training epidemic forecasting models using temporal (time series) data rather than spatiotemporal data. This design choice is motivated by the following considerations:

- We aim to establish the foundation for epidemic pre-training in the temporal setting, which remains unexplored — only one prior work (Kamarthi & Prakash, 2023) addresses this area, and it also focuses on temporal setting. As shown in Sections 4.2-4.5, we address critical questions around generalization, few-shot/zero-shot performance, and pre-training dynamics — topics that remain open even without spatial context.
- Temporal models are broadly applicable and more data-efficient, especially when spatial data is unavailable or unreliable. Many real-world epidemic datasets lack well-defined spatial graphs, and building them (e.g., from mobility or administrative data) is costly and complex, particularly at scale. These inconsistencies also hinder fair comparisons between temporal and spatiotemporal models.
- Our framework is extensible to spatiotemporal modeling. Specifically, the temporal input can be replaced with graph-structured data, and the predictor can incorporate graph-based encoders. Exploring this direction is exciting future work, but we believe temporal pre-training is a crucial first step toward that goal.

**Zero-Shot.** Once pre-trained, our CAPE framework can be directly utilized for zero-shot forecasting, where the model remains frozen and no parameter is updated. Similar to the MOMENT model (Goswami et al., 2024), we retain the pre-trained reconstruction head and mask the last patch of the input to perform forecasting: $\hat{\mathbf{y}} = \hat{\mathbf{x}}_{[T-c:T]}$.

**Data Splits.** For the ILI USA, Measles, and Dengue datasets, we split the data into 60% training, 10% validation, and 30% test. Other datasets are divided into 40% training, 20% validation, and 40% test. During test, we use the model checkpoint with the best validation performance.

**Model Details.** We design our model by stacking 4 layers of the CAPE encoder, each with a hidden size of 512 and 4 attention heads. For compartment representations, we incorporate 16 distinct compartments, each encoded with a size of 512. We constrain two prototypes with monotonic increase or decrease loss, respectively, 6 with the non-monotonic loss, and leave the rest to be unconstrained. To ensure a fair comparison, PatchTST is configured with the same number of layers and hidden size as our CAPE-based model. For all other baseline models, we adopt the architectures as reported in previous studies (Wang et al., 2024a; Kamarthi & Prakash, 2023; Panagopoulos et al., 2021).

**Training Details:**

- We adopt an input length of 36 (Wu et al., 2023; Wang et al., 2024b) and a patch size of 4 for applicable models. For the compartment estimator defined in Eq. equation 1, a shared weight $w_k$ is used for all compartment representations. All results are evaluated using Mean Squared Error (MSE).
- CAPE follows the general EM framework, whose convergence is well-studied (Obata et al., 2024; Qu et al., 2019; Wu, 1983). To ensure stable convergence, we use a small learning rate (1e-5) with L2 regularization, train for 150 epochs, and select the model with the lowest validation error.
- For the training process, we pre-train CAPE and PatchTST on a single Nvidia L40 GPU. During pre-training, we utilize only 70% of the available training data, specifically the first 70% of the dataset for each disease category. We set the learning rate to $1 \times 10^{-5}$. In the CAPE pre-training strategy, we assign a weight of 1e-5 to $\lambda$ to balance the contribution of alignment loss to the whole loss function. We use a general $R_0$ range of 0-20 during pre-training.
- After pre-training, we fine-tune the entire model for five epochs with a changing learning rate, weight decay, and a larger $\lambda$. We also apply the disease-specific $R_0$ (shown in Table 6) for supervision during this phase. The best-performing model is selected based on its performance on the validation set. Similarly, for all baseline models, we train each until convergence and select the optimal model based on validation set performance for the subsequent test.

## A.7 COMPARTMENT INFLUENCE

We also add Gaussian noise to mixture weights (Equation 1), causing performance drops, which highlights the importance of accurate compartment estimation:

Table 7: Model Performance under Different Noise Scales

| Noise Scale (%) | 5% | 10% | 15% | 20% |
|---|---|---|---|---|
| **MSE** | 0.327 | 0.373 | 0.456 | 0.559 |
| **MAPE** | 0.198 | 0.211 | 0.236 | 0.264 |

## A.8 COMPARISONS WITH PEM AND ARIMA

In this section, we provide further comparisons with the Statistical model ARIMA (Panagopoulos et al., 2021), which is configured in an online forecasting setting (Pham et al., 2022; Abdulmajeed et al., 2020) where parameters are updated with each new sample. As shown in Table 8, ARIMA typically outperforms CAPE in the short-term forecasting, while CAPE outperforms ARIMA in long-term forecasting and the averaged performance.

## A.9 ONLINE FORECASTING COMPARISON

We further adopted an online setting used by EINNs Rodríguez et al. (2023) and Epideep Adhikari et al. (2019), where model parameters are consistently updated during forecasting. We compare the results in Table 9.

Table 8: Mean Squared Error (MSE) comparison between ARIMA, CAPE, and PEM models. The lowest MSE for each horizon is marked in bold.

| Horizon | ILI USA | | | ILI Japan | | | Measles | | | Dengue | | | Covid | | |
|---|---|---|---|---|---|---|---|---|---|---|---|---|---|---|---|
| | ARIMA | CAPE | PEM | ARIMA | CAPE | PEM | ARIMA | CAPE | PEM | ARIMA | CAPE | PEM | ARIMA | CAPE | PEM |
| 1 | **0.138** | 0.174 | 0.303 | 0.358 | **0.328** | 0.734 | **0.070** | 0.111 | 0.330 | **0.244** | 0.367 | 0.912 | 33.779 | **25.841** | 36.163 |
| 2 | 0.203 | **0.192** | 0.328 | 0.772 | **0.709** | 0.919 | **0.120** | 0.157 | 0.350 | 0.373 | **0.317** | 0.844 | 33.199 | **25.413** | 29.278 |
| 4 | 0.354 | **0.299** | 0.507 | 1.720 | **1.191** | 1.310 | 0.223 | **0.188** | 0.464 | 0.696 | **0.508** | 1.236 | 32.476 | **24.631** | 33.545 |
| 8 | 0.702 | **0.469** | 0.519 | 2.991 | **1.792** | 1.836 | 0.481 | **0.406** | 0.726 | 1.736 | **1.169** | 1.806 | 36.567 | **33.003** | 39.577 |
| 16 | 1.119 | **0.650** | 0.682 | 2.590 | **1.878** | 1.936 | 1.047 | **0.883** | 1.213 | 4.131 | **2.512** | 2.938 | **42.908** | 49.838 | 49.299 |
| Avg | 0.503 | **0.357** | 0.468 | 1.686 | **1.179** | 1.347 | 0.388 | **0.349** | 0.616 | 1.436 | **0.975** | 1.547 | 35.785 | **31.745** | 37.573 |

Table 9: Comparison of model performance for online evaluation setting.

| Datasets | Horizon | EXPEM | | | EINN | | | EpiDeep | | |
|---|---|---|---|---|---|---|---|---|---|---|
| | | MSE | MAE | RMSE | MSE | MAE | RMSE | MSE | MAE | RMSE |
| ILI_USA | 1 | **32.6287±0.5659** | **5.4137±0.0407** | **5.7119±0.0496** | 34.9396±0.8649 | 5.6884±0.0659 | 5.9105±0.0736 | 46.8892±0.5597 | 6.6289±0.0353 | 6.8474±0.0409 |
| | 2 | 36.7515±0.8805 | **5.7716±0.0959** | 6.0619±0.0724 | **36.5572±1.4472** | 5.8143±0.1435 | **6.0451±0.1192** | 48.3728±0.5039 | 6.7256±0.0351 | 6.9550±0.0363 |
| | 4 | **40.9876±0.5186** | **6.1004±0.0660** | **6.4020±0.0404** | 41.1675±0.3398 | 6.1796±0.0618 | 6.4161±0.0265 | 50.6746±0.1006 | 6.8778±0.0063 | 7.1186±0.0071 |
| Measles | 1 | **0.2042±0.1042** | **0.3971±0.1161** | **0.4364±0.1172** | 0.4718±0.2292 | 0.5851±0.1712 | 0.6688±0.1565 | 1.1160±0.0935 | 0.9424±0.0345 | 1.0555±0.0439 |
| | 2 | **0.2581±0.0495** | **0.4535±0.0538** | **0.5054±0.0513** | 0.4408±0.1457 | 0.5702±0.0988 | 0.6542±0.1132 | 1.0842±0.0513 | 0.9305±0.0227 | 1.0410±0.0248 |
| | 4 | **0.4384±0.0698** | **0.5330±0.0342** | **0.6600±0.0531** | 0.6228±0.0775 | 0.6710±0.0523 | 0.7877±0.0486 | 1.1505±0.0326 | 0.9505±0.0140 | 1.0725±0.0152 |

## A.10 FULL RESULTS ON PRE-TRAIN DATASETS

In addition to evaluating the performance of the models on downstream datasets, we also provide the in-domain evaluation results from the pre-training datasets. Recall that we used 70% data of each disease for pre-training, here we fine-tuned the model on the 70% of each disease and evaluate both CAPE and the pre-trained PatchTST on the rest 30% data. As shown in Table 10, CAPE consistently outperforms PatchTST on 13/15 datasets, proving the effectiveness of our method.

## A.11 FULL RESULTS FOR FEW-SHOT FORECASTING

We present the complete few-shot performance across different horizons in Table 11. While CAPE does not achieve state-of-the-art average performance on the ILI USA dataset with limited training data, it excels in short-term forecasting when the horizon is smaller. Since the authors of PEM (Kamarthi & Prakash, 2023) did not release full code, we implemented the method to the best of our ability based on the paper description.

## A.12 IMPACT OF THE COMPARTMENT DISTRIBUTION AND PRE-TRAIN EPOCHS

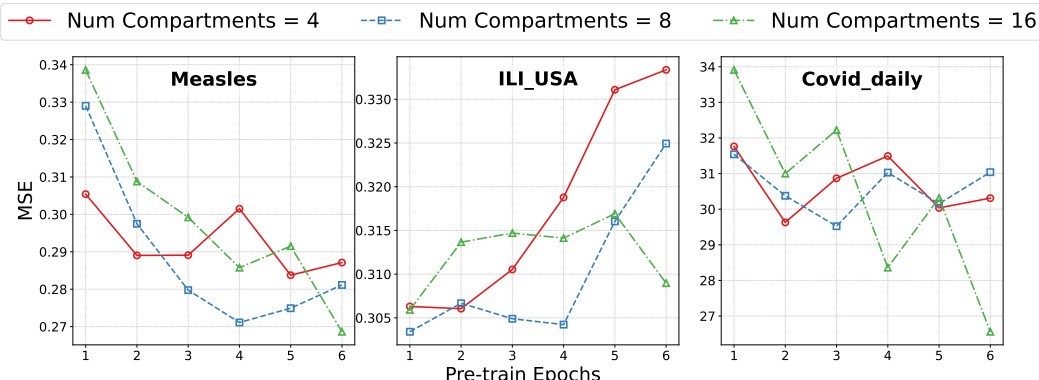

Figure 9: Downstream performance vs. compartment distribution and pre-train epochs.

## A.13 IMPACT OF PRE-TRAINING RATIO ON THE DOWNSTREAM DATASETS

We provide additional evaluations for CAPE on downstream datasets to analyze the impact of the pre-training ratio. As shown in Figure 10, increasing the pre-training ratio eventually improves downstream performance across all datasets.

Table 10: Performance of CAPE and pre-trained PatchTST across diseases in the pre-training datasets. The results presented is the average over horizons of 1,2,4,8,16.

| Disease | Method | Horizon 1 | Horizon 2 | Horizon 4 | Horizon 8 | Horizon 16 | Average |
|---|---|---|---|---|---|---|---|
| Mumps | CAPE | 0.000284 | 0.000290 | 0.000370 | 0.000451 | 0.000539 | **0.000387** |
| | PatchTST | 0.000280 | 0.000310 | 0.000388 | 0.000508 | 0.000627 | **0.000423** |
| Meningococcal Meningitis | CAPE | 0.063022 | 0.066196 | 0.073552 | 0.093547 | 0.108842 | **0.081032** |
| | PatchTST | 0.054611 | 0.061641 | 0.073794 | 0.088404 | 0.096449 | **0.074980** |
| Influenza | CAPE | 0.367677 | 0.510453 | 0.693110 | 0.903920 | 1.037177 | **0.702467** |
| | PatchTST | 0.392925 | 0.644013 | 0.717147 | 0.851498 | 1.061066 | **0.733330** |
| Hepatitis B | CAPE | 0.071834 | 0.072827 | 0.074606 | 0.077816 | 0.068012 | **0.073019** |
| | PatchTST | 0.074016 | 0.082576 | 0.084535 | 0.085867 | 0.074103 | **0.080219** |
| Pneumonia | CAPE | 0.038916 | 0.052092 | 0.082579 | 0.137004 | 0.191675 | **0.100453** |
| | PatchTST | 0.036961 | 0.074596 | 0.096963 | 0.152206 | 0.174871 | **0.107119** |
| Typhoid Fever | CAPE | 0.004918 | 0.004393 | 0.004552 | 0.005051 | 0.005828 | **0.004948** |
| | PatchTST | 0.007068 | 0.005954 | 0.005906 | 0.006519 | 0.006709 | **0.006431** |
| Hepatitis A | CAPE | 0.347792 | 0.349403 | 0.352361 | 0.360705 | 0.315496 | **0.345151** |
| | PatchTST | 0.331339 | 0.349549 | 0.356113 | 0.381637 | 0.338067 | **0.351341** |
| SCAPEet Fever | CAPE | 4.229920 | 5.258288 | 6.787577 | 10.865951 | 13.724634 | **8.173274** |
| | PatchTST | 8.561295 | 13.564009 | 17.241462 | 19.315905 | 20.373520 | **15.811238** |
| Gonorrhea | CAPE | 0.010826 | 0.010900 | 0.011246 | 0.011483 | 0.011898 | **0.011271** |
| | PatchTST | 0.011297 | 0.012223 | 0.013411 | 0.013438 | 0.013241 | **0.012722** |
| Smallpox | CAPE | 0.063829 | 0.065191 | 0.076199 | 0.098973 | 0.157850 | **0.092408** |
| | PatchTST | 0.070972 | 0.076843 | 0.107076 | 0.124042 | 0.165442 | **0.108875** |
| Acute Poliomyelitis | CAPE | 0.254014 | 0.394454 | 0.355898 | 0.480525 | 0.745428 | **0.446064** |
| | PatchTST | 0.094695 | 0.134304 | 0.270908 | 0.392511 | 0.482426 | **0.274969** |
| Diphtheria | CAPE | 0.006789 | 0.005360 | 0.006557 | 0.010682 | 0.014136 | **0.008705** |
| | PatchTST | 0.011019 | 0.008891 | 0.009036 | 0.013048 | 0.015531 | **0.011505** |
| Varicella | CAPE | 0.000119 | 0.000128 | 0.000154 | 0.000212 | 0.000245 | **0.000171** |
| | PatchTST | 0.000109 | 0.000141 | 0.000169 | 0.000237 | 0.000296 | **0.000190** |
| Tuberculosis | CAPE | 0.178741 | 0.170441 | 0.215367 | 0.177671 | 0.198068 | **0.188057** |
| | PatchTST | 0.189156 | 0.209008 | 0.189944 | 0.204680 | 0.277632 | **0.214084** |
| Measle | CAPE | 0.009626 | 0.010982 | 0.016451 | 0.022407 | 0.042980 | **0.020489** |
| | PatchTST | 0.013008 | 0.012608 | 0.020903 | 0.039835 | 0.063844 | **0.030039** |

Table 11: Few-shot learning results with horizons ranging from 1 to 16 future steps. The length of the lookback window is set to 36. Each model is evaluated after being trained on 20%, 40%, 60% and 80% of the full training data.

| Dataset | Horizon | CAPE | | | | | PatchTST | | | | | Dlinear | | | | | MOMENT | | | | | PEM | | | | |
|---|---|---|---|---|---|---|---|---|---|---|---|---|---|---|---|---|---|---|---|---|---|---|---|---|---|---|
| | | 20% | 40% | 60% | 80% | 100% | 20% | 40% | 60% | 80% | 100% | 20% | 40% | 60% | 80% | 100% | 20% | 40% | 60% | 80% | 100% | 20% | 40% | 60% | 80% | 100% |
| ILI USA | 1 | 1.155 | 0.535 | 0.307 | 0.178 | 0.155 | 1.361 | 0.662 | 0.355 | 0.191 | 0.195 | 1.430 | 1.000 | 0.460 | 0.220 | 0.170 | 2.859 | 1.274 | 0.608 | 0.267 | 0.216 | 1.424 | 0.620 | 0.330 | 0.189 | 0.145 |
| | 2 | 1.396 | 0.925 | 0.465 | 0.220 | 0.200 | 1.389 | 0.806 | 0.489 | 0.234 | 0.264 | 2.210 | 1.090 | 0.660 | 0.280 | 0.220 | 3.242 | 1.709 | 0.695 | 0.342 | 0.271 | 1.463 | 0.829 | 0.434 | 0.256 | 0.210 |
| | 4 | 1.770 | 1.154 | 0.640 | 0.306 | 0.270 | 1.923 | 1.215 | 0.656 | 0.387 | 0.385 | 2.500 | 1.670 | 0.720 | 0.380 | 0.310 | 3.910 | 1.901 | 0.891 | 0.399 | 0.356 | 1.889 | 1.186 | 0.625 | 0.393 | 0.312 |
| | 8 | 2.611 | 1.912 | 0.978 | 0.519 | 0.404 | 2.713 | 1.623 | 0.833 | 0.544 | 0.535 | 3.510 | 1.970 | 0.980 | 0.530 | 0.450 | 4.706 | 2.013 | 1.120 | 0.615 | 0.482 | 2.649 | 1.690 | 0.966 | 0.580 | 0.573 |
| | 16 | 3.674 | 2.473 | 1.411 | 0.622 | 0.516 | 3.182 | 1.789 | 1.056 | 0.649 | 0.485 | 4.460 | 2.240 | 1.260 | 0.640 | 0.580 | 5.233 | 2.335 | 1.251 | 0.669 | 0.580 | 3.294 | 1.979 | 1.049 | 0.679 | 0.526 |
| | Avg | **2.121** | **1.400** | **0.760** | **0.369** | **0.309** | 2.114 | 1.219 | 0.677 | 0.401 | 0.373 | 2.822 | 1.594 | 0.816 | 0.412 | 0.346 | 3.990 | 1.847 | 0.913 | 0.459 | 0.381 | 2.143 | 1.261 | 0.681 | 0.419 | 0.353 |
| Dengue | 1 | 3.254 | 1.384 | 0.489 | 0.384 | 0.218 | 3.700 | 1.580 | 0.657 | 0.389 | 0.203 | 3.600 | 1.470 | 0.550 | 0.350 | 0.220 | 4.585 | 2.480 | 0.689 | 0.423 | 0.383 | 3.383 | 1.613 | 0.558 | 0.350 | 0.206 |
| | 2 | 4.463 | 2.340 | 0.735 | 0.487 | 0.301 | 5.832 | 2.159 | 0.846 | 0.507 | 0.296 | 7.090 | 2.170 | 0.820 | 0.510 | 0.310 | 6.609 | 2.990 | 0.922 | 0.587 | 0.521 | 5.404 | 2.257 | 0.869 | 0.507 | 0.300 |
| | 4 | 7.563 | 3.728 | 1.250 | 0.817 | 0.540 | 9.525 | 3.636 | 1.517 | 1.069 | 0.588 | 11.190 | 4.130 | 1.520 | 0.940 | 0.560 | 12.877 | 4.106 | 1.644 | 0.966 | 0.669 | 8.782 | 4.428 | 1.608 | 1.037 | 0.522 |
| | 8 | 15.526 | 7.276 | 2.836 | 1.922 | 1.193 | 19.052 | 9.530 | 3.597 | 2.133 | 1.296 | 21.910 | 9.690 | 3.470 | 2.160 | 1.250 | 23.298 | 9.229 | 3.625 | 2.135 | 1.235 | 17.023 | 8.117 | 3.323 | 2.249 | 1.295 |
| | 16 | 35.870 | 17.204 | 6.469 | 3.946 | 2.210 | 30.451 | 19.616 | 7.238 | 4.289 | 2.536 | 35.350 | 24.640 | 7.890 | 4.780 | 3.060 | 31.115 | 18.877 | 7.200 | 4.551 | 3.984 | 29.934 | 18.861 | 7.368 | 4.390 | 2.497 |
| | Avg | **13.335** | **6.386** | **2.356** | **1.511** | **0.892** | 13.712 | 7.304 | 2.771 | 1.678 | 0.984 | 15.828 | 8.420 | 2.850 | 1.748 | 1.080 | 15.697 | 7.536 | 2.816 | 1.733 | 1.358 | 12.90 | 7.055 | 2.745 | 1.707 | 0.964 |
| Measles | 1 | 0.168 | 0.158 | 0.107 | 0.095 | 0.069 | 0.400 | 0.217 | 0.121 | 0.091 | 0.094 | 0.560 | 0.470 | 0.190 | 0.150 | 0.100 | 1.211 | 0.316 | 0.138 | 0.108 | 0.102 | 0.227 | 0.200 | 0.106 | 0.106 | 0.084 |
| | 2 | 0.229 | 0.256 | 0.165 | 0.134 | 0.096 | 0.511 | 0.325 | 0.186 | 0.148 | 0.127 | 0.680 | 0.400 | 0.320 | 0.220 | 0.150 | 1.376 | 0.367 | 0.159 | 0.167 | 0.138 | 0.313 | 0.339 | 0.155 | 0.153 | 0.127 |
| | 4 | 0.371 | 0.399 | 0.267 | 0.198 | 0.155 | 0.663 | 0.510 | 0.297 | 0.243 | 0.205 | 1.050 | 0.920 | 0.360 | 0.310 | 0.240 | 1.444 | 0.516 | 0.278 | 0.228 | 0.196 | 0.497 | 0.451 | 0.258 | 0.240 | 0.196 |
| | 8 | 0.564 | 0.776 | 0.451 | 0.339 | 0.280 | 1.050 | 1.269 | 0.479 | 0.414 | 0.378 | 1.580 | 1.340 | 0.540 | 0.450 | 0.450 | 1.895 | 1.181 | 0.507 | 0.386 | 0.883 | 0.865 | 1.213 | 0.487 | 0.441 | 0.382 |
| | 16 | 1.086 | 1.408 | 0.917 | 0.658 | 0.743 | 1.692 | 1.847 | 1.157 | 0.900 | 0.723 | 2.100 | 2.520 | 1.480 | 1.170 | 1.030 | 2.379 | 2.192 | 1.041 | 1.468 | 1.183 | 1.448 | 2.275 | 1.145 | 0.880 | 0.740 |
| | Avg | **0.483** | **0.600** | **0.381** | **0.285** | **0.269** | 0.863 | 0.834 | 0.448 | 0.359 | 0.306 | 1.194 | 1.130 | 0.602 | 0.478 | 0.394 | 1.661 | 0.915 | 0.425 | 0.471 | 0.500 | 0.670 | 0.896 | 0.430 | 0.364 | 0.306 |

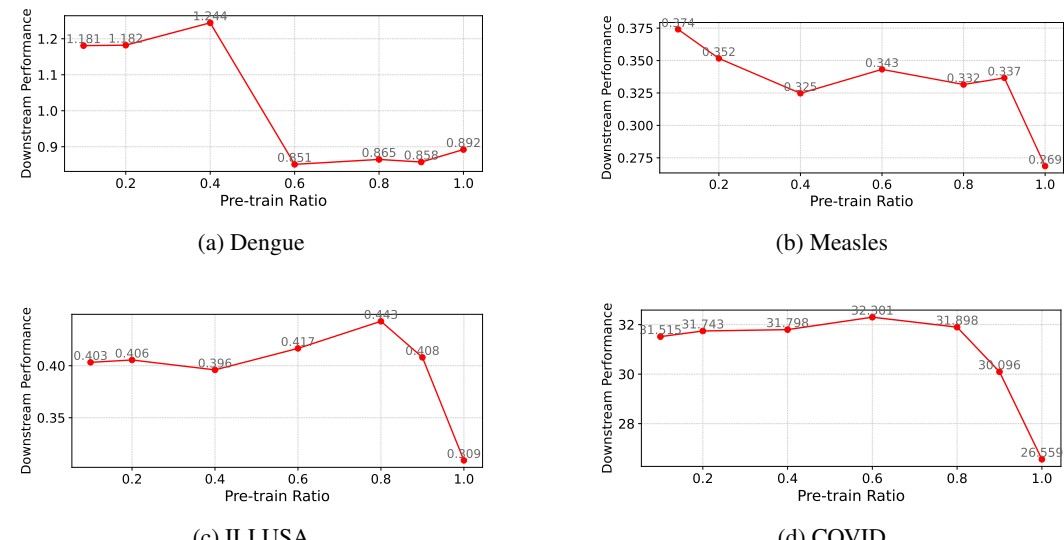

Figure 10: Downstream performance with different ratios of pre-training datasets. The input length is set to 36 and all MSE results are averaged over {1,2,4,8,16} future steps.

## A.14 MITIGATING DISTRIBUTION SHIFT

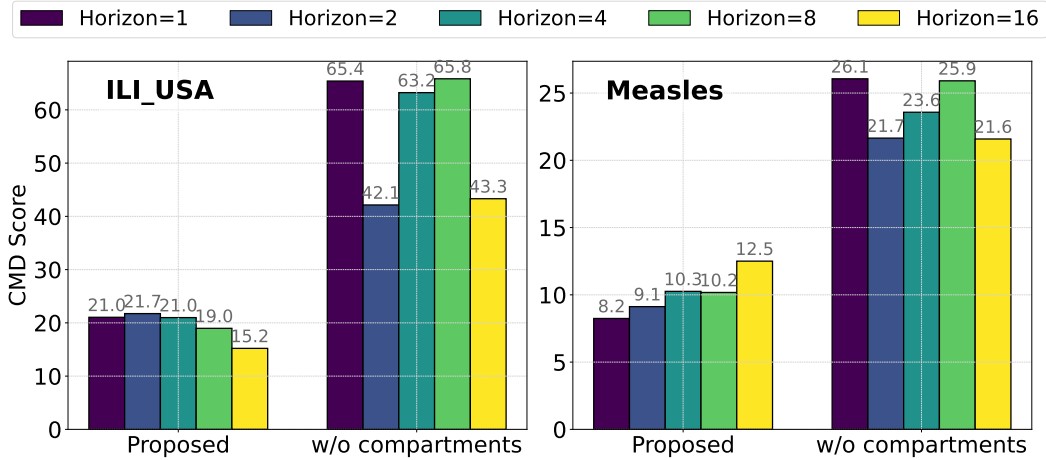

Figure 11: CMD scores w/ and w/o compartment estimate between train/test sets. We measure the CMD scores based on the last embedding output by our model.

## A.15 DISENTANGLING DISEASE-SPECIFIC DYNAMICS

After pre-training, both CAPE and PatchTST were frozen and applied to downstream datasets directly to generate latent space embeddings for each sample. We then employed the Davies-Bouldin Index (DBI) to assess the separability of these embeddings by disease. CAPE consistently achieved lower DBI scores (we show an example in Figure 15), demonstrating a superior ability to distinguish between different diseases in the latent space compared to PatchTST. This enhanced separability highlights the effectiveness of CAPE's compartment estimation and backdoor adjustment. These mechanisms are crucial for mitigating the confounding influence of noisy, spurious, and compartment-dependent factors ($X_{sp}$), thereby enabling the model to better isolate and represent the unique, intrinsic, and causal disease dynamics ($X_{ca}$).

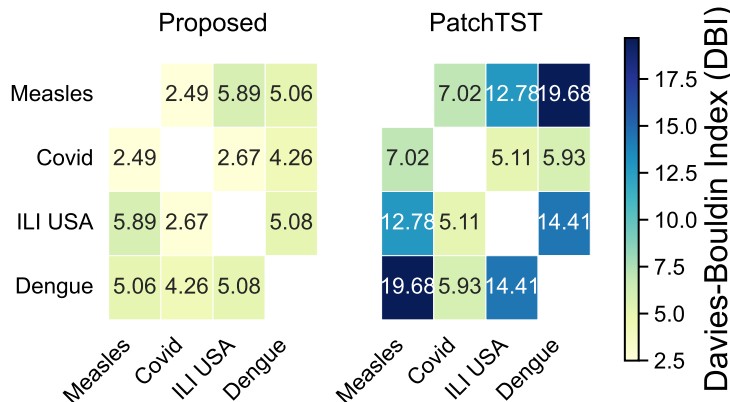

Figure 12: DBI between the embeddings of each pair of downstream datasets from the pre-trained model. (See Figure 15).

## A.16 VISUALIZATION OF THE ESTIMATED COMPARTMENTS

According to $\hat{\mathbf{e}}^{(l)} = \sum_{k=1}^{K} \mathbf{e}_k \pi_k^{(l)}$, an aggregated compartment is the weighted sum of the learned latent compartment representations. Therefore, the estimation shares the same latent space as the fixed representations and we are able to visualize them using t-SNE. As shown in Figure 13, we visualize the aggregated compartments (Estimated) as well as the learned latent compartment (Anchor) from a CAPE model with 8 compartments.

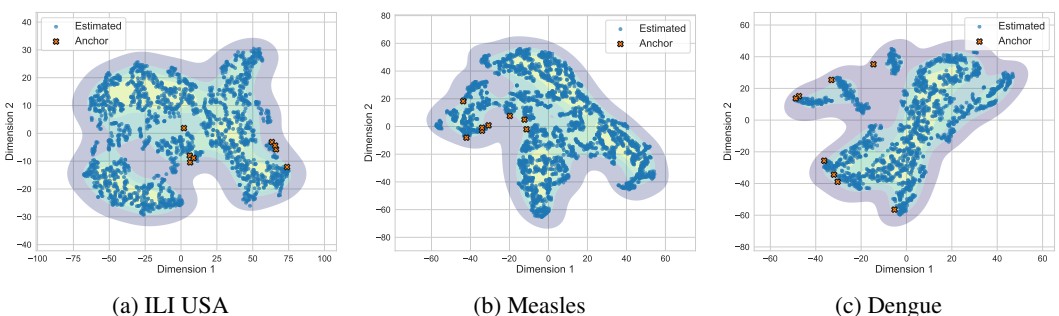

(a) ILI USA         (b) Measles         (c) Dengue

Figure 13: Visualization of the estimated compartment representations using t-SNE.

## A.17 VISUALIZATION OF DISTRIBUTION SHIFT FOR DOWNSTREAM DATASETS

We provide a visualization of the sample distribution used in this study. Each sample has a fixed length of 36, representing the historical infection trajectory. To better understand the distributional differences, we use t-SNE to reduce the data to one dimension and visualize the training and test samples using different colors. As shown in Figure 14, a significant distribution shift is visually apparent across most datasets. To quantitatively assess the distributional differences between the training and test sets, we calculate the Central Moment Discrepancy (CMD) score (Zellinger et al., 2017). The CMD score measures the discrepancy between the central moments of the two distributions up to a specified order $K$. For two distributions $X$ (training set) and $X_{\text{test}}$ (test set), the CMD score is defined as:

$$\text{CMD}(X, X_{\text{test}}) = \|\mu_1(X) - \mu_1(X_{\text{test}})\|_2 + \sum_{k=2}^{K} \|\mu_k(X) - \mu_k(X_{\text{test}})\|_2, \tag{10}$$

where: $\mu_k(X)$ denotes the $k$-th central moment of $X$, defined as: $\mu_k(X) = \mathbb{E}[(X - \mathbb{E}[X])^k]$, and similarly for $\mu_k(X_{\text{test}})$. $\|\cdot\|_2$ is the Euclidean norm. $K$ is the maximum order of moments considered.

The CMD score aggregates the differences in the mean (first moment) and higher-order moments (e.g., variance, skewness), providing a robust measure of the distribution shift. In our experiments, we set $K = 3$ to capture up to the third-order central moments. This score quantitatively complements the visual observations in Figure 14, offering a more comprehensive understanding of the distributional differences between training and test sets.

**Impact of Distribution Shifts.** Distribution shifts between training and test datasets pose significant challenges to the generalizability and robustness of predictive models. When the underlying data distributions differ, models trained on the training set may struggle to maintain their performance on the test set, leading to reduced accuracy and reliability. These discrepancies can arise from various factors, such as temporal changes in infection patterns or geographical variations. In this paper, we assume that the inherent infection pattern of a particular disease remains constant, and the distribution shifts for the disease are primarily caused by the rapidly changing compartment, which results in diverse infection patterns. In the context of epidemic modeling, such shifts are especially critical, as they can undermine the model's ability to accurately predict future infection trends, which is essential for effective public health interventions.

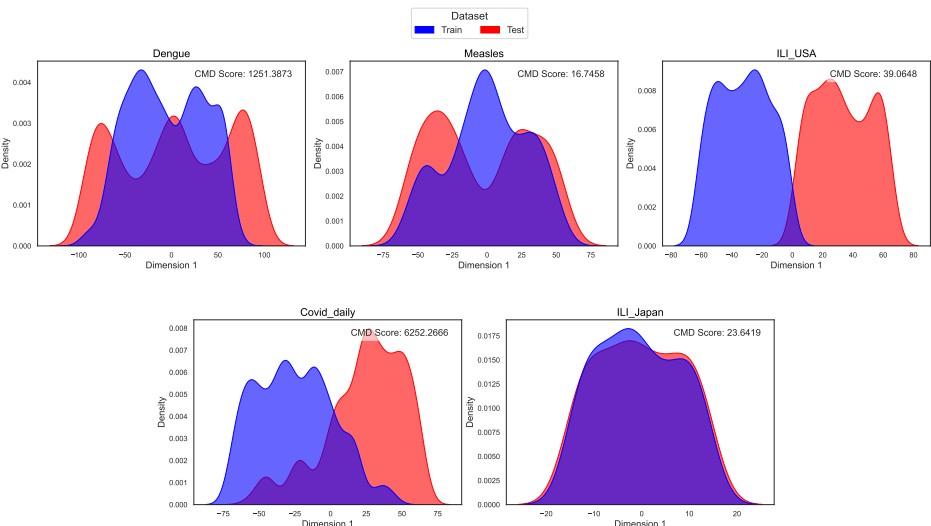

Figure 14: The KDE plot of training set and test set. Each sample contains an infection trajectory of 36 weeks. t-SNE is applied to visualize the distributions of both sets.

### A.18 LATENT SPACE VISUALIZATION OF MEASLE AND COVID DATASETS FROM PRE-TRAINED MODELS.

In order to demonstrate that CAPE effectively disentangles the underlying dynamics of diseases from the influence of the compartment, we visualize the output embeddings for the Measles and COVID datasets by projecting them into a two-dimensional space using t-SNE. Specifically, we utilize the pre-trained model without fine-tuning on these two downstream datasets and visualize $\mathbf{x}^{(L)}$, the final-layer embeddings, as individual data points in the figure. As shown in Figure 15, CAPE (left) visually separates the two datasets more effectively than the pre-trained PatchTST model (right). To quantitatively evaluate the separability of the embeddings, we compute the Davies–Bouldin Index (DBI), which is defined as:

$$\text{DBI} = \frac{1}{K} \sum_{i=1}^{K} \max_{j \neq i} \left( \frac{\sigma_i + \sigma_j}{\|\mu_i - \mu_j\|} \right), \tag{11}$$

where $K$ is the number of clusters (in this case, two: Measles and COVID), $\mu_i$ is the centroid of cluster $i$, $\sigma_i$ is the average intra-cluster distance for cluster $i$, defined as: $\sigma_i = \frac{1}{|C_i|} \sum_{\mathbf{x} \in C_i} \|\mathbf{x} - \mu_i\|$, where $C_i$ is the set of points in cluster $i$, $\|\mu_i - \mu_j\|$ is the Euclidean distance between the centroids

of clusters $i$ and $j$. The DBI measures the ratio of intra-cluster dispersion to inter-cluster separation. Lower DBI values indicate better separability. As shown in Figure 15, CAPE achieves a significantly lower DBI compared to PatchTST, confirming its superior ability to disentangle the underlying disease dynamics from compartmental factors. A more complete result is shown in Figure 12.

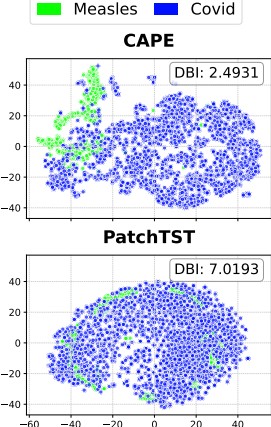

Figure 15: Output latent space of two pre-trained models without fine-tuning from Measle and Covid datasets. Upper: CAPE; Lower: Pre-trained PatchTST.

