# OpenReview forum: "Pre-training Epidemic Time Series Forecasters with Compartmental Prototypes"
_ICLR.cc/2026/Conference — Submitted to ICLR 2026_

### Official Review · Reviewer_offt · 2025-10-31

**Soundness:** 3
**Presentation:** 3
**Contribution:** 3
**Rating:** 6
**Confidence:** 3

**Summary:**

This paper introduces CAPE (Compartmental Prototype Epidemic forecaster), a novel pre-trained model for epidemic time series forecasting. CAPE encodes each outbreak as a mixture of latent compartmental prototypes, capturing hidden epidemiological states (like susceptibility or immunity) and allowing adaptation across diseases and geographies.

**Strengths:**

The introduction of compartmental prototypes provides a flexible and epidemiologically grounded representation of disease progression.

The paper includes robust experiments, ablations, simulation studies, and analysis of distribution shift and representation quality, and CAPE shows reasonable improvements beyond baselines.

**Weaknesses:**

While compartments are learned, their interpretability (in terms of epidemiological meaning) is limited and mostly illustrated by t-SNE and CMD

The optimization involves EM-like updates with contrastive and reconstruction losses, which may hinder scalability in real-time applications.

**Questions:**

Can the authors provide concrete examples of how the learned compartments align with known epidemiological states (e.g., infectious vs. latent)?

Given the iterative EM-style updates and contrastive training, how feasible is this model for real-time epidemic response?

How would CAPE perform on novel disease dynamics that are not well captured in the training distribution (e.g., diseases with complex zoonotic spillovers)?

---

> ### Author Response · Authors · 2025-11-21
>
> We thank the reviewer for the insightful feedback. Here we would like to address the concerns one by one and provide more details in future revisions.
>
> ### ***1. Interpretability of compartments and how the learned compartments align with known epidemiological states.***
>
> We acknowledge that the direct interpretability of latent compartment prototypes is often constrained by the "semantic gap" between the high-dimensional latent space and human-observable concepts, a known limitation in prototype learning [1, 2]. However, certain interpretability can be achieved with the CAPE encoder and appropriate supervision. As shown in Figure 2, CAPE produces compartment *contributions* that align well with classical compartment models like SEIR. While the latent prototypes may not possess a one-to-one semantic mapping to standard epidemiological states (e.g., "Exposed" or "Infectious") in isolation, their aggregate behavior and contribution to the infection count effectively reconstruct valid epidemiological dynamics.
>
> ### ***2. Scalability in real-time forecasting.***
>
> Computational-heavy operations, such as the EM updates with contrastive and reconstruction losses, are exclusively applied during the pre-training stage. The objective of this phase is to internalize the dynamics of historical outbreaks across diverse diseases and locations. Once pre-trained and fine-tuned, CAPE acts as an amortized inference engine, capable of performing real-time online forecasting without the need for repetitive, resource-intensive re-training.
>
> ### ***3. How would CAPE perform on novel disease dynamics outside the training distribution?***
>
> Our experiment on the COVID dataset, which was not included in the pre-training distribution, demonstrates the generalizability of CAPE. While the primary goal of CAPE's pre-training is to expose the model to a vast spectrum of epidemiological trajectories, CAPE also incorporates epidemic priors via epidemic-aware supervision. These priors introduce strong inductive biases that capture fundamental monotonic and non-monotonic disease behaviors. While no model can guarantee perfect adaptation to entirely unprecedented dynamics (e.g., complex zoonotic spillovers), these priors serve as critical regularization, preventing the model from overfitting to the limited data of a specific outbreak and encouraging generalization to plausible epidemiological curves.
>
> ### ***References:***
>
> [1] Nauta M, Van Bree R, Seifert C. Neural prototype trees for interpretable fine-grained image recognition. In Proceedings of the IEEE/CVF conference on computer vision and pattern recognition 2021 (pp. 14933-14943).
>
> [2] Hoffmann A, Fanconi C, Rade R, Kohler J. This looks like that... does it? shortcomings of latent space prototype interpretability in deep networks. arXiv preprint arXiv:2105.02968. 2021 May 5.

---

### Official Review · Reviewer_Sfks · 2025-11-01

**Soundness:** 3
**Presentation:** 3
**Contribution:** 2
**Rating:** 4
**Confidence:** 5

**Summary:**

This work presents a representation learning approach for epidemiological time-series modeling, aiming to improve forecasting performance using observational data. The authors introduce CAPE, an algorithm designed to estimate R0 within an SEIR framework and to enable data augmentation strategies that enhance generalization. Experiments are conducted on diverse and extensive datasets, and the results demonstrate strong performance in zero-shot, few-shot, and transfer settings across multiple benchmarks.

**Strengths:**

1. The problem formulation is well-founded and accurately captures the key challenges in epidemiological modeling.
2. The proposed algorithm for estimating R0 is convincing, supported by a sound theoretical justification. The experiments are comprehensive, with diverse benchmarks and analyses that demonstrate strong generalization.

**Weaknesses:**

1. The work does not provide strong evidence that it effectively addresses non-stationarity in time-series forecasting. The method appears reliable only when the forecasting horizon is shorter than or comparable to the historical window, which does not reflect the harder non-stationary setting.
2. The approach is heavily dependent on the conventional SEIR model and its simulations. If the method relies on such domain-specific priors, it raises the question of why more principled physics-informed models or efficient simulation strategies were not explored. The added complexity of CAPE is not clearly justified.
3. The set of baselines is limited. Several established forecasting methods that incorporate physics-informed modeling or advanced representation learning are missing from the comparison.
4. The paper claims to perform representation learning, but does not provide any analysis demonstrating the quality or usefulness of the learned representations. There is no assessment of feature extraction, embedding structure, or interpretability.

**Questions:**

1. Forecasting horizon remains a key issue. Non-stationarity becomes significant when the prediction window exceeds the length of the historical context. The current experiments use much shorter horizons, making the task more stationary and therefore easier than real-world scenarios.
2. Variation in time-series length during pretraining is not well addressed. In practice, sequences have different lengths, and during evaluation the forecasting horizon must be aligned for inference. The paper provides little clarification on how these mismatches are handled.
3. The representation learning algorithm shows limited advantage. If the learned dynamics are mostly linear (from SEIR ordinary differential equations), simple mix-up could already generate millions of synthetic series for pretraining, which is already proposed. It is unclear why CAPE is necessary or what unique advantage it provides beyond existing mix-up methods. It is also necessary to compare with the work Wang, D., Zhang, S., & Wang, L. (2021, May). Deep epidemiological modeling by black-box knowledge distillation: an accurate deep learning model for COVID-19. In Proceedings of the AAAI Conference on Artificial Intelligence (Vol. 35, No. 17, pp. 15424-15430).

---

> ### Author Response · Authors · 2025-11-21
>
> We thank the reviewer for the insightful feedback. Here we would like to address the concerns one by one and provide more details in future revisions.
>
> ### ***1. Harder non-stationary setting.***
>
> We acknowledge that in the general time series forecasting setting, the horizon can be typically extended very long, e.g, 720 steps. However, in the domain of epidemic forecasting [1, 2, 3], it is a common practice to utilize all the available historical data and perform short-term (1-4 weeks) and long-term (4-8 weeks) forecasting. Therefore, the prediction window is typically smaller than the historical context.
>
> ### ***2. Dependence on the SEIR model.***
>
> We would like to clarify that our method is not constrained by the SEIR model or any single fixed compartmental structure. Unlike traditional models that hard-code specific states (e.g., Susceptible, Exposed, Infected, etc.), CAPE learns a **flexible dictionary of $K$ (e.g., 16) learnable latent prototypes** directly from data. The model infers outbreak dynamics by learning time-varying mixtures of these prototypes, effectively discovering the necessary compartmental structure (whether SIR, SEIR, or complex variants) rather than imposing it *a priori*. The epidemic priors we introduce (monotonic and non-monotonic constraints) act solely as generalized inductive biases to ground these learned prototypes in epidemiological semantics, without restricting them to a specific mathematical formulation.
>
> ### ***3. Comparison with the physics-informed method.***
>
> While the reference research [4] provided by the author is a great example of PINN in epidemic forecasting, their code is not publicly available. Therefore, we provide further experiment here with the EINNs [1], which is an epidemic-informed model guided by the SEIR model.
>
> | **Datasets** | **Horizon** | **EXPEM** |  |  | **EINN** |  |  | **EpiDeep** |  |  |
> | --- | --- | --- | --- | --- | --- | --- | --- | --- | --- | --- |
> |  |  | **MSE** | **MAE** | **RMSE** | **MSE** | **MAE** | **RMSE** | **MSE** | **MAE** | **RMSE** |
> | **ILI_USA** | **1** | **32.6287±0.5659** | **5.4137±0.0407** | **5.7119±0.0496** | 34.9396±0.8649 | 5.6884±0.0659 | 5.9105±0.0736 | 46.8892±0.5597 | 6.6289±0.0353 | 6.8474±0.0409 |
> |  | **2** | 36.7515±0.8805 | **5.7716±0.0959** | 6.0619±0.0724 | **36.5572±1.4472** | 5.8143±0.1435 | **6.0451±0.1192** | 48.3728±0.5039 | 6.7256±0.0351 | 6.9550±0.0363 |
> |  | **4** | **40.9876±0.5186** | **6.1004±0.0660** | **6.4020±0.0404** | 41.1675±0.3398 | 6.1796±0.0618 | 6.4161±0.0265 | 50.6746±0.1006 | 6.8778±0.0063 | 7.1186±0.0071 |
> | **Measles** | **1** | **0.2042±0.1042** | **0.3971±0.1161** | **0.4364±0.1172** | 0.4718±0.2292 | 0.5851±0.1712 | 0.6688±0.1565 | 1.1160±0.0935 | 0.9424±0.0345 | 1.0555±0.0439 |
> |  | **2** | **0.2581±0.0495** | **0.4535±0.0538** | **0.5054±0.0513** | 0.4408±0.1457 | 0.5702±0.0988 | 0.6542±0.1132 | 1.0842±0.0513 | 0.9305±0.0227 | 1.0410±0.0248 |
> |  | **4** | **0.4384±0.0698** | **0.5330±0.0342** | **0.6600±0.0531** | 0.6228±0.0775 | 0.6710±0.0523 | 0.7877±0.0486 | 1.1505±0.0326 | 0.9505±0.0140 | 1.0725±0.0152 |
>
> ### ***4. Representation learning.***
>
> We acknowledge that the direct interpretability of latent compartment prototypes is constrained by the "semantic gap" between the high-dimensional latent space and human-observable concepts, a known limitation in prototype learning [5, 6]. However, the CAPE encoder, when paired with appropriate supervision, bridges this gap to achieve functional interpretability. Specifically, the model is able to: (1) produce contribution scores that align with established compartment models, as shown in Figure 2; and (2) generate diverse combinatorial patterns of latent compartments, as illustrated in Figure 13.
>
> ### ***5. Variation in time-series length during pre-training.***
>
> We acknowledge that time series lengths vary between pre-training and downstream evaluation. In this work, we follow existing evaluation standard in epidemic forecasting and use a fixed historical context window length for all models during pre-training and fine-tuning. To address the task difference between these stages, we provide a short description in Section 4.3. Overall, we achieve this by swapping the prediction head. Our pre-training dataset is composed of cleaned, consecutive weekly infection trajectories. Each trajectory is sampled via a sliding window, and SSL methods with a pre-training head are applied to each sample. Finally, during evaluation on downstream datasets, we switch to a different forecasting head; this head is fine-tuned to predict future cases, effectively aligning with the target forecasting horizon.

---

> > ### Author Response · Authors · 2025-11-21
> >
> > ### ***References:***
> >
> > [1] Rodríguez A, Cui J, Ramakrishnan N, Adhikari B, Prakash BA. Einns: epidemiologically-informed neural networks. In Proceedings of the AAAI conference on artificial intelligence 2023 Jun 26 (Vol. 37, No. 12, pp. 14453-14460).
> >
> > [2] Panagopoulos G, Nikolentzos G, Vazirgiannis M. Transfer graph neural networks for pandemic forecasting. In Proceedings of the AAAI Conference on Artificial Intelligence 2021 May 18 (Vol. 35, No. 6, pp. 4838-4845).
> >
> > [3] Adhikari B, Xu X, Ramakrishnan N, Prakash BA. Epideep: Exploiting embeddings for epidemic forecasting. In Proceedings of the 25th ACM SIGKDD international conference on knowledge discovery & data mining 2019 Jul 25 (pp. 577-586).
> >
> > [4] Wang, D., Zhang, S., & Wang, L. (2021, May). Deep epidemiological modeling by black-box knowledge distillation: an accurate deep learning model for COVID-19. In Proceedings of the AAAI Conference on Artificial Intelligence (Vol. 35, No. 17, pp. 15424-15430).
> >
> > [5] Nauta M, Van Bree R, Seifert C. Neural prototype trees for interpretable fine-grained image recognition. In Proceedings of the IEEE/CVF conference on computer vision and pattern recognition 2021 (pp. 14933-14943).
> >
> > [6] Hoffmann A, Fanconi C, Rade R, Kohler J. This looks like that... does it? shortcomings of latent space prototype interpretability in deep networks. arXiv preprint arXiv:2105.02968. 2021 May 5.

---

### Official Review · Reviewer_7dLt · 2025-11-01

**Soundness:** 2
**Presentation:** 3
**Contribution:** 1
**Rating:** 2
**Confidence:** 4

**Summary:**

This paper aims to learn an epidemiological foundation model for predicting the spread of infectious diseases. It proposes a pre-training framework for epidemic forecasting models that integrates mechanistic guidance through the Next Generation Matrix (NGM) and a contrastive learning objective to improve generalization across diseases and regions.

**Strengths:**

(1) The adaptation of the Next Generation Matrix (NGM) method to improve the deep learning model looks interesting.

(2) There has been a non-trivial effort to engage with the literature in mathematical epidemiology, which is commendable.

(3) Paper presentation is good.

**Weaknesses:**

(1) The paper presents ideas and theoretical expressions that have already appeared in prior work but fails to cite the corresponding sources, which could mislead a less careful reader into thinking they are novel contributions. For example, the proof of the lower and upper bounds of the Next Generation Matrix (NGM) method is not original – it is a standard linear-algebra result applied to the classical NGM formulation of R_0. However, since the proof is provided without any citation, readers may assume it is the authors’ contribution. What appears to be new in this paper is how those bounds are used as a differentiable regularizer for R_0 derived from the model’s learned latent compartments, but this distinction should be made explicit.

(2) A similar issue arises with the idea of using neural networks to map to compartments in an epidemiological model, which has been explored in several previous studies [1]–[4], all of which are omitted from citation. In particular, the monotonicity loss introduced by the authors is strikingly similar to that proposed in [2], yet no reference is provided.

(3) I believe the only genuine technical contribution of this paper lies in the integration of the NGM framework. The remaining components are minor adaptations of well-known techniques that offer limited new insight: for instance, the use of contrastive learning guided by the mechanistic model to determine positive and negative samples. Unfortunately, the ablation results indicate that both the NGM component and the contrastive learning loss contribute only marginal improvements. Overall, this paper can be seen as a nice synthesis of several existing approaches, presenting ideas that may appear novel but are, in fact, extensions of prior work — often without the proper citations that would clarify this connection.

(4) The experimental setup appears to have potential flaws. In epidemic forecasting, it is standard practice to evaluate model performance in an online forecasting setting to better reflect real-world conditions. However, the authors apply this setup only in the appendix (Section A.8), while all main results rely on an offline evaluation. This raises concerns about the practical utility of the proposed method, particularly since ARIMA outperforms it in short-term forecasting (1 to 4 weeks), the most common use case in real-world applications (see [5]). It is also unclear why the authors chose to focus on 1- to 16-week forecasting horizons for the main experiments, while 1-4 is the most relevant.

[1] Wang, L., Adiga, A., Chen, J., Sadilek, A., Venkatramanan, S. and Marathe, M., 2022, June. Causalgnn: Causal-based graph neural networks for spatio-temporal epidemic forecasting. In Proceedings of the AAAI conference on artificial intelligence (Vol. 36, No. 11, pp. 12191-12199).

[2] Rodríguez, A., Cui, J., Ramakrishnan, N., Adhikari, B. and Prakash, B.A., 2023, June. Einns: epidemiologically-informed neural networks. In Proceedings of the AAAI conference on artificial intelligence (Vol. 37, No. 12, pp. 14453-14460).

[3] Kargas, N., Qian, C., Sidiropoulos, N.D., Xiao, C., Glass, L.M. and Sun, J., 2021, May. Stelar: Spatio-temporal tensor factorization with latent epidemiological regularization. In Proceedings of the AAAI conference on artificial intelligence (Vol. 35, No. 6, pp. 4830-4837).

[4] Gao, J., Sharma, R., Qian, C., Glass, L.M., Spaeder, J., Romberg, J., Sun, J. and Xiao, C., 2021. STAN: spatio-temporal attention network for pandemic prediction using real-world evidence. Journal of the American Medical Informatics Association, 28(4), pp.733-743.

[5] Cramer, E.Y., Ray, E.L., Lopez, V.K., Bracher, J., Brennen, A., Castro Rivadeneira, A.J., Gerding, A., Gneiting, T., House, K.H., Huang, Y. and Jayawardena, D., 2022. Evaluation of individual and ensemble probabilistic forecasts of COVID-19 mortality in the United States. Proceedings of the National Academy of Sciences, 119(15), p.e2113561119.

**Questions:**

Following weakness (2): The manuscript references several deep learning approaches for epidemic forecasting ([6]–[8]); however, it remains unclear why the authors consider it unnecessary to include direct comparisons with these methods (e.g., in their main results, table 1). In particular, for [6], the authors state that corresponding results are provided in the appendix, yet these results are not present. Furthermore, the omission of works [1]–[4], which address closely related ideas, further limits the completeness of the evaluation. The authors are encouraged to clarify the rationale for excluding these baselines from conceptual discussion and/or empirical comparison.

Following weakness (2): The papers CausalGNN [1] and EINNs [2] present ideas that appear closely related to those introduced in the current manuscript. In those works, the neural network predicts the population in each compartment of a mechanistic model, and the authors of EINNs also introduce a monotonicity loss similar to the one described in your Equation (4). Could the authors elaborate on the key differences between their proposed model and these approaches? Additionally, it would be important to clarify why this prior work was omitted from the discussion of related literature.

[6] Adhikari, B., Xu, X., Ramakrishnan, N., and Prakash, B. A. Epideep: Exploiting embeddings for epidemic forecasting. In Proceedings of the 25th ACM SIGKDD international conference on knowledge discovery & data mining, pp. 577–586, 2019.

[7] Panagopoulos, G., Nikolentzos, G., and Vazirgiannis, M. Transfer graph neural networks for pandemic forecasting. In Proceedings of the AAAI Conference on Artificial Intelligence, volume 35, pp. 4838–4845, 2021.

[8] Kamarthi, H. and Prakash, B. A. Pems: Pre-trained epidmic time-series models. arXiv preprint arXiv:2311.07841, 2023.

---

> ### Author Response · Authors · 2025-11-21
>
> We thank the reviewer for the insightful feedbacks and we apologize for missing some references, which is not intentional. Here we would like to address the concerns one by one and provide more details, including the references, in the future revisions.
>
> ### ***1. Novelty for the lower and upper bounds.***
>
> We would like to clarify the novelty and contribution of the proposed bounds from two perspectives:
>
> (1) While we agree that the mathematical tools used (spectral radius bounds via matrix norms) are fundamental in linear algebra, this specific formulation of lower and upper bounds in the context of epidemic forecasting has not been used to the best of our knowledge. Since the reviewer did not provide any evidence of our proposed bounds appearing in prior work, we would like to argue that this formulation in the epidemic context is still novel.
>
> (2) While efforts to find bounds for $R_0$ or epidemic dynamics exist in the other literature [1,2], our formulation differs in both mathematical structure and computational purpose. Specifically, Diekmann et al. [2] define $R_0$ as the spectral radius $\rho(FV^{-1})$3. However, calculating the spectral radius requires the eigen decomposition of the product of $F$ and the inverse of $V$. In a deep learning context, computing the matrix inverse $V^{-1}$ and subsequent eigenvalues at every training step is computationally expensive and numerically unstable, often leading to gradient explosions. **In contrast**, the bounds we provided ($\frac{\sigma_{min}(F)}{\sigma_{max}(V)} \le R_0 \le \frac{\sigma_{max}(F)}{\sigma_{min}(V)}$) are designed specifically to **avoid matrix inversion**. By constraining the singular values of $F$ and $V$ separately, we provide a stable, differentiable path for the gradient-based optimizer. This allows the model to be regularized by $R_0$ constraints efficiently, which is a distinct contribution from the analytical bounds found in prior literature.

---

> > ### Author Response · Authors · 2025-11-21
> >
> > ### ***2. Synthesis of several existing approaches.***
> >
> > We use several well-known strategies for pre-training, including contrastive learning and random masking, which we do not claim as our central contribution. The major contribution of our paper lies in three perspectives: 1. exploration of pre-training in epidemic forecasting, 2. latent compartment prototypes learning, and 3. derivation of epidemic-aware regularizations.
> >
> > ### ***3. Marginal improvement.***
> >
> > We respectfully ask the reviewer to consider our proposed components in greater detail, particularly the pre-training strategy and epidemic-aware supervision. These elements contribute notable performance gains: an average improvement of 10.75% from pre-training and 5.8% from epidemic-aware regularization, highlighting their effectiveness.
> >
> > ### ***4. Using neural networks to map to compartments in an epidemiological model and methodological differences with CausalGNN and EINNs.***
> >
> > We acknowledge that combining neural networks with epidemiological models is an active research area and we apologize for missing relevant citations, which we have included in the revised version. However, CAPE differs fundamentally from frameworks like EINNs and CausalGNN by moving from "calibrating a fixed structure" to "discovering a flexible structure." While EINNs and CausalGNN map NNs to **pre-defined, rigid ODEs**, CAPE learns a **flexible dictionary of latent prototypes** directly from data.
> >
> > (1) **Fixed vs. Learned Structure (Addressing Heterogeneity):**
> >
> > ◦ **EINNs [3]** and **CausalGNN [4]** require the user to hard-code the compartmental graph *a priori* (e.g., SEIRM for COVID-19 in EINNs or SIRD in CausalGNN). The neural networks are restricted to inferring parameters or states for these specific equations.
> >
> >  ◦ **CAPE** does not impose a fixed transition graph. Instead, it initializes learnable embeddings (**compartmental prototypes**) and learns to express outbreak dynamics as a time-varying mixture of these latent states333. This allows CAPE to handle **structural heterogeneity** (e.g., SIR vs. SEIR) automatically without manual re-specification.
> >
> > (2) **Explicit Solving vs. Latent Representation:**
> >
> > ◦ **EINNs** and **CausalGNN** explicitly solve differential equations or match gradients to estimate specific counts (S, I, R) or parameters ($\beta, \gamma$).
> >
> > ◦ **CAPE** operates in a **latent space**, avoiding the brittleness of explicit state estimation. We use **Epidemic-Aware Regularization** (monotonicity and $R_0$ constraints) to guide learned prototypes toward epidemiological semantics via soft constraints rather than hard ODE solvers.
> >
> > (3) **Single-Task vs. Foundation Model:**
> >
> > ◦ **EINNs** and **CausalGNN** are trained as "specialists" on single datasets to fit specific curves.
> >
> > ◦ **CAPE is a pre-trained foundation model.** By learning a vocabulary of prototypes across 17 diseases, it captures universal dynamics, enabling the robust zero-shot and few-shot transfer demonstrated in our results. For instance, CAPE outperforms strong baselines by up to **26.06%** in zero-shot forecasting on unseen diseases (e.g., RSV) and achieves a **10.3% lower average MSE** in few-shot settings across diverse datasets.
> >
> > **Summary:**
> > In short, while EINNs and CausalGNN are powerful tools for *parameter estimation* on known structures, CAPE is designed for *generalizable forecasting* across unknown or heterogeneous dynamics. By learning latent compartmental prototypes rather than enforcing fixed ODEs, CAPE achieves superior flexibility and transferability, as evidenced by our state-of-the-art performance in zero-shot and few-shot benchmarks.

---

> ### Author Response · Authors · 2025-11-21
>
> ### ***5. Monotonic loss function.***
>
> While we acknowledge the conceptual similarity, as both methods adapt the monotonicity loss from Muralidhar et al. [5], the **target and architectural role of the loss differ significantly** in our framework:
>
> (1) **Target of Constraint:** EINNs applies the loss to the **explicit output states** (specifically the $S$ and $R$ predictions) to ensure the final forecast trajectory is physically valid. In contrast, CAPE applies the loss to the **internal latent mixture weights** ($\pi$) of the learned prototypes.
>
> (2) **Architectural Goal:** In EINNs, the goal is to enforce constraints on the solution of the differential equation. In CAPE, the goal is **representation disentanglement**. We use monotonicity to guide specific *latent* prototypes to acquire epidemiological semantics (e.g., behaving *like* a Susceptible compartment). This disentanglement allows the model to learn a transferable "vocabulary" of dynamics, achieving a better performance compared to EINNs on both ILI_USA and Measles forecasting, as shown in the experiment in Q7.
>
> ### ***6. Comparison with epidemic models.***
>
> We apologize for missing the comparison with the PEM model in the appendix, which is an unintentional mistake, and the experiment has been added in the following table. Our paper focuses on univariate time series forecasting, which does not involve spatial/graph modeling; thus, graph-based models like [6, 7] are excluded.
>
> | Horizon | ILI USA |  |  | ILI Japan |  |  | Measles |  |  | Dengue |  |  | Covid |  |  |
> | --- | --- | --- | --- | --- | --- | --- | --- | --- | --- | --- | --- | --- | --- | --- | --- |
> |  | ARIMA | CAPE | PEM | ARIMA | CAPE | PEM | ARIMA | CAPE | PEM | ARIMA | CAPE | PEM | ARIMA | CAPE | PEM |
> | 1 | 0.138 | 0.174 | 0.303 | 0.358 | 0.328 | 0.734 | 0.07 | 0.111 | 0.33 | 0.244 | 0.367 | 0.912 | 33.779 | 25.841 | 36.163 |
> | 2 | 0.203 | 0.192 | 0.328 | 0.772 | 0.709 | 0.919 | 0.12 | 0.157 | 0.35 | 0.373 | 0.317 | 0.844 | 33.199 | 25.413 | 29.278 |
> | 4 | 0.354 | 0.299 | 0.507 | 1.72 | 1.191 | 1.31 | 0.223 | 0.188 | 0.464 | 0.696 | 0.508 | 1.236 | 32.476 | 24.631 | 33.545 |
> | 8 | 0.702 | 0.469 | 0.519 | 2.991 | 1.792 | 1.836 | 0.481 | 0.406 | 0.726 | 1.736 | 1.169 | 1.806 | 36.567 | 33.003 | 39.577 |
> | 16 | 1.119 | 0.65 | 0.682 | 2.59 | 1.878 | 1.936 | 1.047 | 0.883 | 1.213 | 4.131 | 2.512 | 2.938 | 42.908 | 49.838 | 49.299 |
> | Avg | 0.503 | **0.357** | 0.468 | 1.686 | **1.179** | 1.347 | 0.388 | **0.349** | 0.616 | 1.436 | **0.975** | 1.547 | 35.785 | **31.745** | 37.573 |

---

> > ### Author Response · Authors · 2025-11-21
> >
> > ### ***7. Online forecasting setup and horizon selection.***
> >
> > In terms of the selection of forecasting horizons, we chose both short-term forecasting settings (1-4 weeks) and long-term forecasting settings (beyond 4 weeks). Such separation is also adopted in other literature, like the EINNs paper.
> >
> > In terms of online/real-time forecasting setup, we follow standard protocols in the related literature [6, 7, 8], and partition the epidemic dataset chronologically into train, validation, and test sets to strictly avoid temporal data leakage. In our evaluation, the model performs real-time forecasting using fixed parameters derived from the training phase. This differs from the "rolling retraining" approach found in other studies [3, 9], where the model is re-optimized on the entire history prior to every forecasting instance. While the retraining method offers potential adaptability, it is computationally intensive. Notably, a comprehensive benchmark comparing the efficacy of fixed-parameter versus rolling-retraining evaluation protocols remains absent in current literature. Here, we would like to provide additional experiments on the real-time setting adopted in EINN [2] and compare with epidemic models, including EINN and EpiDeep.
> >
> > | **Datasets** | **Horizon** | **EXPEM** |  |  | **EINN** |  |  | **EpiDeep** |  |  |
> > | --- | --- | --- | --- | --- | --- | --- | --- | --- | --- | --- |
> > |  |  | **MSE** | **MAE** | **RMSE** | **MSE** | **MAE** | **RMSE** | **MSE** | **MAE** | **RMSE** |
> > | **ILI_USA** | **1** | **32.6287±0.5659** | **5.4137±0.0407** | **5.7119±0.0496** | 34.9396±0.8649 | 5.6884±0.0659 | 5.9105±0.0736 | 46.8892±0.5597 | 6.6289±0.0353 | 6.8474±0.0409 |
> > |  | **2** | 36.7515±0.8805 | **5.7716±0.0959** | 6.0619±0.0724 | **36.5572±1.4472** | 5.8143±0.1435 | **6.0451±0.1192** | 48.3728±0.5039 | 6.7256±0.0351 | 6.9550±0.0363 |
> > |  | **4** | **40.9876±0.5186** | **6.1004±0.0660** | **6.4020±0.0404** | 41.1675±0.3398 | 6.1796±0.0618 | 6.4161±0.0265 | 50.6746±0.1006 | 6.8778±0.0063 | 7.1186±0.0071 |
> > | **Measles** | **1** | **0.2042±0.1042** | **0.3971±0.1161** | **0.4364±0.1172** | 0.4718±0.2292 | 0.5851±0.1712 | 0.6688±0.1565 | 1.1160±0.0935 | 0.9424±0.0345 | 1.0555±0.0439 |
> > |  | **2** | **0.2581±0.0495** | **0.4535±0.0538** | **0.5054±0.0513** | 0.4408±0.1457 | 0.5702±0.0988 | 0.6542±0.1132 | 1.0842±0.0513 | 0.9305±0.0227 | 1.0410±0.0248 |
> > |  | **4** | **0.4384±0.0698** | **0.5330±0.0342** | **0.6600±0.0531** | 0.6228±0.0775 | 0.6710±0.0523 | 0.7877±0.0486 | 1.1505±0.0326 | 0.9505±0.0140 | 1.0725±0.0152 |
> >
> > ### ***References:***
> >
> > [1] Avram F, Adenane R, Basnarkov L, Johnston MD. Algorithmic approach for a unique definition of the next-generation matrix. Mathematics. 2023 Dec 21;12(1):27.
> >
> > [2] Diekmann O, Heesterbeek JA, Roberts MG. The construction of next-generation matrices for compartmental epidemic models. Journal of the royal society interface. 2010 Jun 6;7(47):873-85.
> >
> > [3] Rodríguez A, Cui J, Ramakrishnan N, Adhikari B, Prakash BA. Einns: epidemiologically-informed neural networks. InProceedings of the AAAI conference on artificial intelligence 2023 Jun 26 (Vol. 37, No. 12, pp. 14453-14460).
> >
> > [4] Wang L, Adiga A, Chen J, Sadilek A, Venkatramanan S, Marathe M. Causalgnn: Causal-based graph neural networks for spatio-temporal epidemic forecasting. InProceedings of the AAAI conference on artificial intelligence 2022 Jun 28 (Vol. 36, No. 11, pp. 12191-12199).
> >
> > [5] Muralidhar N, Islam MR, Marwah M, Karpatne A, Ramakrishnan N. Incorporating prior domain knowledge into deep neural networks. In2018 IEEE international conference on big data (big data) 2018 Dec 10 (pp. 36-45). IEEE.
> >
> > [6] Panagopoulos G, Nikolentzos G, Vazirgiannis M. Transfer graph neural networks for pandemic forecasting. InProceedings of the AAAI Conference on Artificial Intelligence 2021 May 18 (Vol. 35, No. 6, pp. 4838-4845).
> >
> > [7] Gao J, Sharma R, Qian C, Glass LM, Spaeder J, Romberg J, Sun J, Xiao C. STAN: spatio-temporal attention network for pandemic prediction using real-world evidence. Journal of the American Medical Informatics Association. 2021 Apr 1;28(4):733-43.
> >
> > [8] Du H, Zhao Y, Zhao J, Xu S, Lin X, Chen Y, Gardner LM, Yang HF. Advancing real-time infectious disease forecasting using large language models. Nature Computational Science. 2025 Jun 6:1-4.
> >
> > [9] Adhikari B, Xu X, Ramakrishnan N, Prakash BA. Epideep: Exploiting embeddings for epidemic forecasting. InProceedings of the 25th ACM SIGKDD international conference on knowledge discovery & data mining 2019 Jul 25 (pp. 577-586).

---

### Official Review · Reviewer_skHA · 2025-11-02

**Soundness:** 2
**Presentation:** 3
**Contribution:** 2
**Rating:** 4
**Confidence:** 3

**Summary:**

The paper proposes a pre-trained framework for epidemic forecasting that learns transferable epidemic dynamics from diverse historical disease data. CAPE models outbreaks as mixtures of latent population states that it discovers directly from surveillance data. They use self-supervised objectives: masked time-series reconstruction and contrastive learning, and epidemic-aware regularizers. CAPE captures robust, interpretable representations that generalize across diseases and regions. Experiments on 17 diseases across 50+ regions show that CAPE outperforms baseline time-series and epidemic models in zero-shot, few-shot forecasting.

**Strengths:**

1. The motivation for a pre-trained model for epidemic forecasting is important
2. Epidemic regularized used in pre-trained models in novel
3, Results on provided set of diseases is significant and are compared against without pre-trained models

**Weaknesses:**

1. Comparison with SOTA epidemic models are lacking (https://arxiv.org/abs/2207.09370 has a few examples)
2. The benchmarks can be expanded with more datasets from wide range of diseases. Eg: see datasets such as (https://www.tycho.pitt.edu/)
3. Are the datasets evaluated on real-time forecasting setup? What is the training validation split. Real-time forecasting setup is more realistic for ongoing epidemics where such models are more useful.
4. Can the model provide probabilistic forecasts? This is very crucial for real-world deployment.

**Questions:**

See Weaknesses

---

> ### Author Response · Authors · 2025-11-21
>
> We thank the reviewer for the insightful feedback. Here we would like to address the concerns one by one and provide more details in future revisions.
>
> ### ***1. Lacking comparison with SOTA epidemic models and the online forecast setting.***
>
> The implementation of online/real-time forecasting setting vary across different prior studies and we follow a commonly adopted formulation established in earlier research. Specifically, we partition the epidemic dataset chronologically into train, validation, and test sets to strictly avoid temporal data leakage. In our evaluation, the model performs real-time forecasting using fixed parameters derived from the training phase, consistent with standard protocols in the literature [1, 2, 3]. This differs from the "rolling retraining" approach found in other studies [4, 5], where the model is re-optimized on the entire history prior to every forecasting instance. While the retraining method offers potential adaptability, it is computationally intensive. Notably, a comprehensive benchmark comparing the efficacy of fixed-parameter versus rolling-retraining evaluation protocols remains absent in current literature.
>
> Together with the requested comparison of SOTA epidemic models, we would like to provide additional experiments on the rolling retraining setting with two SOTA models: EINN [4] and EpiDeep [5], which are provided in https://arxiv.org/abs/2207.09370.
>
> | **Datasets** | **Horizon** | **EXPEM** |  |  | **EINN** |  |  | **EpiDeep** |  |  |
> | --- | --- | --- | --- | --- | --- | --- | --- | --- | --- | --- |
> |  |  | **MSE** | **MAE** | **RMSE** | **MSE** | **MAE** | **RMSE** | **MSE** | **MAE** | **RMSE** |
> | **ILI_USA** | **1** | **32.6287±0.5659** | **5.4137±0.0407** | **5.7119±0.0496** | 34.9396±0.8649 | 5.6884±0.0659 | 5.9105±0.0736 | 46.8892±0.5597 | 6.6289±0.0353 | 6.8474±0.0409 |
> |  | **2** | 36.7515±0.8805 | **5.7716±0.0959** | 6.0619±0.0724 | **36.5572±1.4472** | 5.8143±0.1435 | **6.0451±0.1192** | 48.3728±0.5039 | 6.7256±0.0351 | 6.9550±0.0363 |
> |  | **4** | **40.9876±0.5186** | **6.1004±0.0660** | **6.4020±0.0404** |41.1675±0.3398 |6.1796±0.0618 |6.4161±0.0265 | 50.6746±0.1006 | 6.8778±0.0063 | 7.1186±0.0071 |
> | **Measles** | **1** | **0.2042±0.1042** | **0.3971±0.1161** |**0.4364±0.1172** | 0.4718±0.2292 | 0.5851±0.1712 | 0.6688±0.1565 | 1.1160±0.0935 | 0.9424±0.0345 | 1.0555±0.0439 |
> |  | **2** |**0.2581±0.0495** |**0.4535±0.0538** |**0.5054±0.0513** |0.4408±0.1457 |0.5702±0.0988 |0.6542±0.1132 | 1.0842±0.0513|0.9305±0.0227|1.0410±0.0248|
> |  | **4** | **0.4384±0.0698** | **0.5330±0.0342** | **0.6600±0.0531** | 0.6228±0.0775 | 0.6710±0.0523 | 0.7877±0.0486 | 1.1505±0.0326 | 0.9505±0.0140 | 1.0725±0.0152 |
>
> ### ***2. The benchmarks can be expanded with more datasets from a wide range of diseases. Eg: see datasets such as (https://www.tycho.pitt.edu/)***
>
> We agree that expanding the number of datasets would be valuable. **Our work already incorporates data from Project Tycho**, and we selected a diverse set of 17 diseases for pre-training, each providing infection trajectories from multiple geographic locations. A central aim of our study is to demonstrate the utility of pre-training for epidemic forecasting, as shown in Figures 3, 4, and 6.
>
> ### ***3. Can the model provide probabilistic forecasts?***
>
> We agree that probabilistic forecasts are essential for real-world epidemic forecasting. In this work, however, our primary objective is to demonstrate the value of pre-training for improving forecasting performance, rather than to develop a full probabilistic forecasting framework. Although our model does not include a dedicated uncertainty-estimation component at this moment, it is compatible with model-agnostic approaches such as conformal prediction [6], which can be applied on top of our deterministic outputs to provide uncertainty quantification.
>
> ### ***References:***
>
> [1] Du H, Zhao Y, Zhao J, Xu S, Lin X, Chen Y, Gardner LM, Yang HF. Advancing real-time infectious disease forecasting using large language models. Nature Computational Science.
>
> [2] Panagopoulos G, Nikolentzos G, Vazirgiannis M. Transfer graph neural networks for pandemic forecasting. In Proceedings of the AAAI Conference on Artificial Intelligence.
>
> [3] Gao J, Sharma R, Qian C, Glass LM, Spaeder J, Romberg J, Sun J, Xiao C. STAN: spatio-temporal attention network for pandemic prediction using real-world evidence. Journal of the American Medical Informatics Association.
>
> [4] Rodríguez A, Cui J, Ramakrishnan N, Adhikari B, Prakash BA. Einns: epidemiologically-informed neural networks. In Proceedings of the AAAI conference on artificial intelligence.
>
> [5] Adhikari B, Xu X, Ramakrishnan N, Prakash BA. Epideep: Exploiting embeddings for epidemic forecasting. In Proceedings of the 25th ACM SIGKDD international conference on knowledge discovery & data mining.
>
> [6] Stankeviciute K, M Alaa A, Van der Schaar M. Conformal time-series forecasting. Advances in neural information processing systems.

---

### Meta-Review · Area_Chair_sZKr · 2025-12-30

**Summary:**

I find this submission to be a competent but ultimately incremental contribution to epidemic forecasting. The core proposition—pre-training a transformer on diverse historical disease surveillance data using compartmental prototypes—is sensible, and the extensive benchmarking across 17 diseases and multiple evaluation settings (zero-shot, few-shot, full-shot) is valuable. However, after carefully weighing reviewer concerns and the author rebuttal, several issues remain unresolved that preclude acceptance.

The novelty claims are weaker than presented. Reviewer 7dLt correctly identifies that the monotonicity loss closely resembles that in EINNs (Rodríguez et al., 2023), and the Next Generation Matrix bounds are standard spectral radius inequalities applied to a known epidemiological formulation.

The distinction between CAPE's "flexible structure" and prior work's "fixed structure" is real but incremental—both approaches use neural networks to predict compartment-related quantities. The ablation results further undermine the paper's positioning: removing epidemic regularization costs only 2-5% performance while removing pre-training costs ~10%, suggesting the main value lies in generic pre-training rather than the proposed epidemic-specific components.

Additionally, ARIMA remains competitive at short horizons (1-4 weeks), which is the most operationally relevant regime for public health response, and the lack of probabilistic forecasting capability is a significant limitation for deployment.

**Reviewer Concerns:**

Reviewer 7dLt raised concerns about missing citations to EINNs, CausalGNN, and related work, as well as the similarity of the monotonicity loss to prior formulations. The authors acknowledged the citation omissions and provided additional references. They argued that CAPE differs because it applies monotonicity to latent mixture weights rather than explicit compartment states, but I find this distinction to be incremental rather than fundamental.

Reviewer 7dLt also questioned why ARIMA outperforms CAPE at short horizons (1-4 weeks). The rebuttal confirmed this pattern in Table 8, arguing that CAPE wins on average and at longer horizons. This response is technically accurate but concerning from a practical standpoint, as short-term forecasting is most actionable for outbreak response.

Reviewers skHA and Sfks requested comparisons with epidemic-specific methods (EINNs, EpiDeep). The authors provided Table 9 showing CAPE outperforms these methods on ILI USA and Measles in an online evaluation setting, which partially addresses this concern. The request for probabilistic forecasts (skHA) was acknowledged but not addressed; the authors suggested conformal prediction as a post-hoc solution, which does not resolve the limitation.

**Reviewer Scores:**

Reviewer skHA: 4 → 5. The additional comparisons with EINNs/EpiDeep address their main concerns, and they indicated openness to acceptance.

Reviewer 7dLt: 2 → 3. The rebuttal provides useful clarifications but does not resolve the core concerns about incremental novelty and marginal ablation gains from epidemic components.

Reviewer Sfks: 4 → 4. Clarifications were provided but do not fundamentally change the assessment.

Reviewer ofit: 6 → 6. The positive review remains; concerns about interpretability were acknowledged but not fully resolved.

---

### Decision · Program_Chairs · 2026-01-26

Reject